



# Calibration and Field Testing of Cavity Ring-Down Laser Spectrometers Measuring CH$_4$, CO$_2$, and δ$^{13}$CH$_4$ Deployed on Towers in the Marcellus Shale Region

Natasha L. Miles[1], Douglas K. Martins[1,2], Scott J. Richardson[1], Christopher W. Rella[3], Caleb Arata[3,4], Thomas Lauvaux[1], Kenneth J. Davis[1], Zachary R. Barkley[1], Kathryn McKain[5], Colm Sweeney[5]

[1]Department of Meteorology and Atmospheric Science, The Pennsylvania State University, University Park, Pennsylvania, 16802, USA
[2]FLIR Systems, Inc, West Lafayette, Indiana, 47906, USA (current affiliation)
[3]Picarro, Inc., Santa Clara, California, 95054, USA
[4]University of California, Berkeley, California, 94720, USA
[5]National Oceanic and Atmospheric Administration / University of Colorado, Boulder, Colorado, 80305, USA

*Correspondence to*: N. L. Miles (nmiles@psu.edu)

**Abstract**. Four in-situ cavity ring-down spectrometers (G2132-i, Picarro, Inc.) measuring methane dry mole fraction (CH$_4$), carbon dioxide dry mole fraction (CO$_2$) and the isotopic ratio of methane (δ$^{13}$CH$_4$) were deployed at four towers in the Marcellus Shale natural gas extraction region of Pennsylvania. The calibration of the continuous isotopic methane analyzers used in this study required both a linear calibration and a mole fraction correction, and a correction for cross-interference from ethane. In this paper, we describe laboratory and field calibration of the analyzers for tower-based applications, and characterize their performance in the field for the period January – December 2016. Prior to deployment, each analyzer was calibrated using high methane mole fraction air bottles with various isotopic ratios, from biogenic to thermogenic source values, diluted in zero air. Furthermore, at each tower location, three field calibration tanks were employed, from ambient to high mole fractions, with various isotopic ratios. By testing multiple calibration schemes, we determined an optimized field calibration method. A method to correct for cross interference from ethane is also described. Using an independent field tank for evaluation, the standard deviation of 4-hour means of the isotopic ratio of methane difference from the known value was found to be 0.26 ‰ δ$^{13}$CH$_4$. Following improvements in the field calibration tank sampling scheme, the standard deviation of 4-hour means was 0.11 ‰, well within the target compatibility of 0.2 ‰. Round robin style testing using tanks with near ambient isotopic ratios indicated mean errors of −0.14 to 0.03 ‰ for each of the analyzers. Flask to in-situ comparisons showed mean differences over the year of 0.02 and 0.08 ‰, for the East and South towers, respectively.

Regional sources in this region were difficult to differentiate from strong perturbations in the background. During the afternoon hours, the median enhancements of isotopic ratio measured at three of the towers, compared to the background tower, were −0.15 to 0.12 ‰ with standard deviations of the 10-min isotopic ratio enhancements of 0.8



‰. In terms of source attribution, analyzer compatibility of 0.2 ‰ $\delta^{13}CH_4$ affords the ability to distinguish a 50 ppb
$CH_4$ peak from a biogenic source from one originating from a thermogenic source. Using a Keeling plot approach
for the non-afternoon data at a tower in the center of the study region, we determined the source isotopic signature to
be –31.2 ‰, consistent with a deep-layer Marcellus natural gas source.



## 1 Introduction

Quantification of regional greenhouse gas emissions resulting from natural gas extraction activities is critical for determining the climate effects of natural gas usage compared to coal or oil. Studies have shown that the emission rates as a percentage of production vary significantly from reservoir to reservoir. An aircraft-based mass balance study in the Uintah basin in Utah (Karion et al., 2013) found a methane emission rate of 6.2–11.7 % of production, exceeding the 3.2 % threshold for natural gas climate benefits compared to coal determined by Alvarez et al. (2012). In the Denver-Julesburg basin in Colorado, Pétron et al. (2014) found an emissions rate of 4 % of production, again using an aircraft mass balance approach. The Barnett Shale, one of the largest production basins in the United States with 8 % of total U.S. natural gas production, was found to exhibit a lower emission rate of 1.3–1.9 % (Karion et al., 2015). Using a model optimization approach for aircraft data, Barkley et al. (2017) found the weighted mean emission rate from unconventional natural gas production and gathering facilities in the Marcellus region in northeastern Pennsylvania, a region with mostly dry natural gas, to be 0.36 % of total gas production.

Differentiating $CH_4$ emissions from natural gas activities from other sources (e.g., wetlands, cattle, landfills) is key to documenting the greenhouse gas impact of natural gas production and to evaluate the effectiveness of emissions reduction activities. The isotopic ratio of methane ($\delta^{13}CH_4$) is particularly useful in this regard (Coleman et al., 1995). In general, heavy isotope ratios are characteristic of thermogenic $CH_4$ sources (i.e., fossil-fuel based) and light isotope ratios are characteristic of biogenic sources (Dlugokencky et al., 2011). Schwiezke et al. (2016) compiled a comprehensive database of isotopic methane source signatures, indicating signatures of –44.0 ‰ for globally averaged fossil-fuel sources of methane, –62.2 ‰ for globally averaged microbial sources such as wetlands, ruminants, and landfills, and –22.2 ‰ for globally averaged biomass burning sources. Atmospheric measurements of $\delta^{13}CH_4$ have been used to partition emissions of $CH_4$ into source categories (e.g., Mikaloff Fletcher et al., 2004a,b; Kai et al., 2011). It is important to note, however, that for fossil-fuel sources of methane, isotopic ratios of methane vary significantly from reservoir to reservoir (e.g., Townsend-Small et al., 2015; Rella et al., 2015), and with depth in a single reservoir (Molofsky et al.,2011; Baldassare et al., 2014).

The isotopic ratio of methane has traditionally been measured with continuous flow gas chromatography/ isotope ratio mass spectrometry, with repeatability of ±0.05‰ (Fisher et al., 2006). Cavity ring-down spectroscopy (CRDS) analyzers measuring continuous isotopic ratio of methane in addition to methane and carbon dioxide (Rella et al., 2015), have recently become commercially available, are more robust in the field, and have the potential to aid with source attribution of the methane signals. CRDS is a laser-based technique in which the infrared absorption loss caused by a gas in the sample cell is measured to quantify the mole fraction of the gas. The analyzers utilize three highly reflective mirrors such that the flow cell has an effective optical path length of 15–20 km, allowing highly precise measurements. The temperature and pressure of the sample cell is tightly controlled, improving the stability of the measurements (Crosson 2008). Rella et al. (2015) describe the operation of CRDS (Picarro, Inc., model G2132-i) analyzers, including cross-interference from other gases, and general calibration approach.



Aircraft-based studies cover large areas, but the temporal coverage is limited. Tower-based networks offer a
complementary approach, making continuous measurements over long periods of time. At the Boulder Atmospheric
Observatory (BAO) tall tower, daily flask measurements are found to contain enhanced levels of methane and other
alkanes, compared to the other tall towers in the National Oceanic and Atmospheric Administration (NOAA)
network (Pétron et al., 2012). Tower measurements allow for continuous measurements in the well mixed boundary
layer which are influenced by both nearby sources and the integrated effect of the upstream emissions. While towers
provide near continuous coverage of regional emissions, specific emissions sources with specific isotopic signatures
are often diluted by mixing, making the enhancements above background very small.
Rella et al. (2015) described the use of two tanks to correct for analyzer drift of the isotopic ratio measured by the
G2132-i analyzers. In this approach, the variables of interest, i.e., the total methane mole fraction and the isotopic
ratio, are directly calibrated. The drift terms in the calibration equations have differing dependence on mole
fraction, requiring the use of at least two tanks for calibration. For this study, three field calibration tanks were
deployed at each tower location, two for the field calibration and one as an independent test.
An alternative calibration approach is to calibrate the isotopologues of methane (i.e., the $^{13}CH_4$ and $^{12}CH_4$ mole
fractions), instead of the total methane and the isotopic ratio. This approach has been applied to Fourier Transform
infrared and isotope ratio infrared spectrometers measuring $\delta^{13}C$ and $\delta^{18}O$ of $CO_2$ in air (Griffith et al., 2012; Wen
et al., 2013; Flores et al., 2017).
Here we focus on the specific application of tower-based measurements. We describe laboratory calibration of the
G2132-i analyzers, optimized field calibration strategy, calibration results, and comparison to flasks. Our first
objective is determining how best to calibrate the isotopic measurements in a field setting, given the field calibration
tanks that were deployed for the network. We address several questions related to the effects of applying the
calibration using different combinations of the tanks. The second objective is to determine the compatibility
achieved for the isotopic measurements in the current field deployment, using round-robin style testing and
comparisons to flasks as our primary metrics. We also evaluate the performance of the G2132-i analyzers in terms
of $CH_4$ and $CO_2$ mole fractions measurements.
**2 Compatibility goals**
In this paper, we describe a network of four tower-based atmospheric observation locations, measuring $CH_4$ and
$CO_2$ dry mole fractions and $\delta^{13}CH_4$ using CRDS (Picarro, Inc., model G2132-i) analyzers in the Marcellus shale
region in north-central Pennsylvania. Because this is the first network of isotopic ratio of methane analyzers to date,
the needed compatibility has not yet been defined. Thus, our compatibility goals for $CO_2$ and $CH_4$ mole fractions
follow the WMO compatibility recommendation for global studies: 0.1 ppm for $CO_2$ (in the Northern Hemisphere)



and 2 ppb for $CH_4$ (GAW Report No. 229, 2016). Here we use the term compatibility, as advised in the GAW
Report No. 229 (2016), to describe the difference between two measurements, rather than the absolute accuracy of
those measurements.
For $\delta^{13}CH_4$, we set our target compatibility, somewhat arbitrarily, at 0.2 ‰, thought to be a reasonable goal based on
laboratory testing prior to deployment and the results shown in Rella et al. (2015). This goal corresponds to the
WMO extended compatibility goal for the isotopic ratio of methane, which was deemed sufficient for regionally
focused studies with large local fluxes. The measured signal at the towers is a mixture of the source and the
background (Pataki et al., 2003), and the ability to distinguish between a biogenic and thermogenic source depends
on the difference of the source isotopic signature from background and the peak strength in terms of methane mole
fraction. Equating the slope of a source and the background with the slope of a mixture and the background on a
Keeling plot (Keeling, 1961), the measured isotopic ratio enhancement is given by
$$\Delta\delta = (\delta_{src} - \delta_{back})\frac{\Delta CH_4}{CH_{4,meas}} \qquad\qquad (1)$$
This equation is represented graphically in Fig. 1. If there are two possible sources in a region, a biogenic source at
–60 ‰ and a thermogenic source at –35 ‰, for example, the difference in isotopic ratio enhancement is at least
three times the compatibility goal of 0.2 ‰ (and thus distinguishable) for a peak strength of 50 ppb $CH_4$ or greater,
assuming a measured $CH_4$ mole fraction of 2000 ppb and a background isotopic ratio of –47 ‰. In this case, the
biogenic source would measure 0.3 ‰ above the background, as opposed to the thermogenic source measuring 0.3
‰ below the background. As shown in Fig. 1, sources closer to the background in isotopic ratio require a larger
peak in $CH_4$ and those further from the background can be attributed with a smaller peak in $CH_4$.
**3 Methods: Laboratory testing**
**3.1 Allan standard deviation testing**
Allan standard deviation testing (Allan, 1966) is a useful tool for testing the noise and drift response of
instrumentation. The Allan standard deviation for each averaging interval is proportional to the range of values for
each averaging interval. This range typically decreases for increasing averaging interval, as the noise is averaged
out. Thus, the results are critical for defining the minimum averaging time required for a given target compatibility.
As the averaging interval increases, however, analyzer drift may contribute, placing an upper bound on the optimal
averaging interval.
To calculate the Allan standard deviation of the G2132-i analyzers used in this study, one tank containing an
ambient mole fraction of $CH_4$ (1.9 ppm), and $CO_2$ (~400 ppm) mole fraction and one tank containing high mole



fraction of CH$_4$ (9.7 ppm) and an ambient mole fraction of CO$_2$ (~400 ppm) were sampled with an analyzer for 24
hours. For simplicity, we call these the "high" and "low" tanks, respectively, and they are described further in
Section 4.2. We tested both as the noise is known to be less for higher mole fractions, and at least one tank with
higher CH$_4$ mole fraction is necessary for the isotopic ratio calibration.
**3.2 Analyzer laboratory calibration**
**3.2.1 Experimental set-up**
Prior to field deployment, each analyzer was calibrated for CH$_4$ and CO$_2$ mole fraction. Four NOAA-calibrated
tertiary standards (traceable to the WMO X2004 scale for CH$_4$ and the WMO X2007 scale for CO$_2$) were used for
the linear mole fraction calibration, as described in Richardson et al. (2017). These NOAA tertiary standards ranged
between 1790 and 2350 ppb CH$_4$, and between 360 and 450 ppm CO$_2$.
To calibrate the $\delta^{13}$CH$_4$ measurement prior to deployment, four different target mixing ratios, each at four different
known isotopic ratios were sampled by the four analyzers using the experimental setup in Fig. 2. Commercially-
available isotopic standard bottles (Isometric Instruments, Inc., product numbers L-iso1, B-iso1, T-iso1 and H-iso1)
were diluted with zero air to produce mixtures with varying CH$_4$ mixing ratios and $\delta^{13}$CH$_4$. The gravimetrically-
determined zero air (Scott Marrin, Inc.) was natural ultra-pure air, containing no methane or other alkanes but
ambient levels of CO$_2$. The isotopic calibration standard bottles each contained approximately 2500 ppm of CH$_4$ at
–23.9, –38.3, –54.5, and –66.5 ‰ $\delta^{13}$CH$_4$, with uncertainty of ±0.2 ‰ reported by the supplier. Mass flow-
controllers (MC-1SCCM and MC-500SCCM, Alicat Scientific, Inc.) and a 6-port rotary valve (EUTA-2SD6MWE,
Valco Instruments Co., Inc.) were used to direct the standard bottle air for each isotopic calibration standard bottle
into a mixing volume (~4 m of 1/8 in, 0.32 cm OD stainless steel tubing; TSS285-120F, VICI Precision Sampling,
Inc.) at 0.400 sccm and mixed with zero CH$_4$ air at 137, 161, 303, and 555 sccm to create target CH$_4$ mole fractions
of 7.3, 6.2, 3.3, and 1.8 ppm, respectively. Thus 16 CH$_4$ mole fraction/isotopic ratio pairs were produced. The
accuracy of the mass flow controllers can be a significant source of error in making mixtures. Here the nominal
range of the mass flow controllers was 1 sccm and 500 sccm, respectively, and the accuracy was ±0.2 % of full
scale. To avoid isotopic fractionation at the head of the low-flow mass flow controller, the flow of the zero air was
varied rather than the isotope standard. It is possible that fractionation did occur due to the tees used to direct gas
into the individual analyzers. For this reason, it would have been preferable to set up the analyzers to sample from a
common mixing volume.
A 3-way solenoid valve (091-0094-900, Parker Hannifin Corp.) was used just downstream of the mixing volume to
stop flow from the zero air tank and Isometric Instrument bottles and allow flow from the working standards. In this
way, the working standards were sequentially calibrated. The results of these calibrations are described in Section
4.2 and these working standards are referred to as the "high" and "low" field calibration tanks.




The first mixture of each isotopic standard was sampled for 60 minutes to flush out the span gas line and to avoid
isotopic fractionation at the head of the span mass flow controller.  Subsequent dilutions using the same isotopic
standard were sampled for 20 minutes each and each dilution was repeated twice.  Observations were collected at
~0.5 Hz and the final 5 minutes of data for each dilution were averaged to compare against the target value.  We
note that these laboratory tests were completed prior to the Allan standard deviation testing and that the averaging
times were not sufficient to achieve the desired compatibility for all but the highest mole fractions.  Thus only the
highest mole fractions are used for the linear calibration of the analyzers.
**3.2.2 Application of calibration equations**
The first step in the calibration process is to remove the nearly linear error that is a function of isotopic ratio. We
applied methods leading from the theoretical framework developed by Rella et al. (2015) to calibrate the isotopic
ratio data.  Applying a linear fit to highest mole fraction values (7.3 ppm) for known $\delta^{13}CH_4$ values (–23.9, –38.3, –
54.5, –66.5 ‰) for each analyzer, we determined the linear calibration coefficients $p_1$ and $p_0$.
$$[\delta^{13}CH_4]_{intermediate} = p_1[\delta^{13}CH_4]_{measured} + p_0. \qquad (2)$$
For this step, we used only the highest mole fraction values because $\delta^{13}CH_4$ is more precise for higher mole
fractions, as is apparent from the Allan standard deviation tests (Section 6.1).
To correct for the $CH_4$ mole fraction dependence of the measured $\delta^{13}CH_4$, the two time-dependent drift parameters
described in Rella et al. (2015) $c_0$ and $\chi$ must be determined.  Here $c_0$ varies because of spectral variations in the
optical loss of the empty cavity and $\chi$ varies because of errors in the temperature or pressure of the gas, or changes
in the wavelength calibration.  These parameters are defined in Eq. (15) of Rella et al. (2015).  A coefficient
describing the changes in the crosstalk between the two methane isotopologues was ignored, following Rella et al.
(2015).  We determined $c_0$ and $\chi$ using measurements at –23.9 ‰ for a high mole fraction (7.3 ppm) and a low mole
fraction (1.8 ppm).  We then applied Eq. (12) of Rella et al. (2015)
$$[\delta^{13}CH_4]_{calibrated} = [\delta^{13}CH_4]_{intermediate} + \frac{c_0}{c_{12}} + \chi([\delta^{13}CH_4]_{intermediate} - B), \qquad (3)$$
to correct for the $CH_4$ mole fraction dependence of $\delta^{13}CH_4$.  Here $c_{12}$ is the measured $[^{12}CH_4]$ and
$$B = p_1 B_{default} + p_0, \qquad (4)$$
with $B_{default}$ being –1053.59 ‰.  We followed Rella et al. (2015) and ignored the contribution of an additional offset
term that depends on neither mole fraction nor isotopic ratio.  Note that the slope of the linear calibration was the
only component of the calibration that was not adjusted in the field using field calibration tanks (Section 6.3).
**4 Methods: Field deployment**
**4.1 Study area**
Four CRDS isotopic $CH_4$ analyzers (G2132-i, Picarro, Inc.) were deployed on commercial towers 46–61 m AGL in
northeast Pennsylvania (Fig. 3). The South and North towers were located on the southern and northern edges of the
unconventional gas well region, respectively, and were intended to measure background values depending on the
wind direction. Winds in the study area were most often from the west (Fig. 4). Measurements began in May 2015,
but a complete set of field tanks necessary for calibration of $\delta^{13}CH_4$ was not deployed until January 2016. This
study thus describes results from the period January – December 2016. In addition to the continuous G2132-i
analyzers, the East and South towers were also equipped with NOAA flask sampling systems (Turnbull et al. 2012),
measuring a suite of > 55 gases (including greenhouse gases, hydrocarbons, and halocarbons) and $\delta^{13}CH_4$.
**4.2 In-situ field calibration tanks**
At each tower site, three field calibration tanks were utilized, as listed in Table 1. We anticipated using one tank,
calibrated by the National Oceanic and Atmospheric Administration (NOAA) for $CH_4$ and $CO_2$ mole fractions and
by the Institute of Arctic and Alpine Research (INSTAAR) for $\delta^{13}CH_4$, for the $CH_4$ and $CO_2$ mole fraction
calibration and as an independent check of the $\delta^{13}CH_4$ calibration, and using two additional tanks for the $\delta^{13}CH_4$
calibration. Since no long-term field calibration of isotopic measurements have yet been demonstrated, we
experimented with different ways to use these three tanks and present two optimized scheme. This analysis also
includes suggestions for improving this approach in future field deployments. We first describe the preparation of
the field calibration tanks, then the method used to test their application to field measurements.
One tank was filled and calibrated by NOAA/INSTAAR. This tank was sampled quasi-daily and used to adjust the
intercept for the $CH_4$ and $CO_2$ mole fraction calibrations (Richardson et al., 2017). The constituents of this tank
were at typical ambient levels (as listed in Table 1), and for the purposes of this paper, we call it the "target".
Two additional tanks were sampled at the tower sites (Table 1). These tanks were filled using ultra-pure air and
spiked (using Isometric Instruments, Inc bottles) by Scott Marrin, LLC, (one at 1.9–2.1 ppm $CH_4$ and –23.9
‰ $\delta^{13}CH_4$) and one at 9.7–10.5 ppm $CH_4$ at –38.3 ‰ $\delta^{13}CH_4$). Recall that these are called the "low" and "high"
tanks, for simplicity. These tanks contained ambient levels of $CO_2$ (368 – 407 ppm). They were calibrated for
$\delta^{13}CH_4$ in the laboratory prior to deployment by first applying a linear calibration for $\delta^{13}CH_4$ using measurements
from each of four Isometric Instruments bottles (–23.9, –38.3, –54.5, –66.5 ‰), diluted with zero air to 10.3 – 10.4
ppm $CH_4$. Then a mole fraction correction was applied using the –23.9‰ bottle diluted to 10.4 ppm $CH_4$ and the –
38.3 ‰ bottle diluted to 1.9 ppm. These calibration results are shown in Table 1. The values assigned to the tanks
differed slightly (with the differences ranging in magnitude from 0.01 to 0.38 ‰) from the bottles used for spiking.



Possible reasons for these slight differences include noise in the measurement, fractionation upon tank-filling, and
bottle assignment error with the 0.2 ‰ uncertainty reported by the supplier (Isometric Instruments, Inc.).
The sampling scheme and procedure for using these field calibration tanks at each tower to correct the ambient
$\delta^{13}CH_4$ measurements is described in Sections 3.2.2 and 6.2.
**4.3 In-situ field calibration gas sampling system**
The flow diagram of the field calibration system is shown in Fig. 5. Polyethylene/aluminum composite tubing (¼
in, 0.64 cm OD, Synflex 1300, Eaton Corp.) was used to sample from the top of each tower for the CRDS analyzer
and a separate sample line made from ⅜ in (0.95 cm) OD Synflex 1300 tubing was used for the flask sampling
packages. The top end of each tube was equipped with a rain shield to prevent liquid water from entering the
sampling line. Separate lines were used for the CRDS and flask sampling lines because of the large flow rates
required for the flask samples (Turnbull et. al., 2012) and to ensure independence of the CRDS and flask
measurements.
For the continuous in-situ measurement system, switching between sample and calibration gases was accomplished
using a 6-port rotary valve (EUTA-2SD6MWE, Valco Instruments Co, Inc.). Stainless steel tubing (1/8 in, 0.32 cm
OD, TSS285-120F, VICI Precision Sampling, Inc.) and single-stage regulators (Y11-C444B590, Airgas, Inc.) were
used for sampling the field calibration tanks. A Nafion dryer (MD-070-96S-2, PermaPure) in the reflux
configuration, using an additional pump (ME1, Vacuubrand, Inc.) on the outlet of the Nafion dryer (Fig. 5), was
used to dry the sample air (~0.06 % $H_2O$) and humidify the calibration gases (0.02 % $H_2O$). The $CH_4$ mole fraction
was corrected for water vapor following Rella et al. (2015 supp), and the $CO_2$ mole fraction following Chen et al.

22  (2010).

A cycle including 90 min of ambient sampling, 6 min sampling the high mole fraction field calibration tank, and 10
min sampling the low mole fraction field calibration tank was repeated 12 times, then the target tank was sampled
for 10 min (occurring every ~21 hours, to test for diurnal effects). Thus, there were 13.5 calibration cycles for the
high and low tanks each day, on average. The first 4 min of data were discarded each time after switching gases to
ensure sufficient flushing of the sample cell. Note that the Allan standard deviation results (Section 6.1) indicate
that sampling for 4 min for the high tank and for 32 min for the low and target tanks is required to achieve our target
compatibility of 0.2 ‰ $\delta^{13}CH_4$. Thus, this averaging time was achieved in one calibration cycle for the high tank,
but in 5.3 calibration cycles for the low and target tanks (completed in about 10 hours in the case of the sampling
scheme utilized for most of the deployment). An improved sampling strategy was implemented on 3 December
2016 and is discussed in Section 6.3.



**4.4 Method for determination of optimized application of the field calibration gases**

The calibration procedure described in Section 3.2.2 was applied to the in-situ data. The optimized strategy for application of the field calibration gases in terms of which tanks to use and how often to apply the calibration was determined via testing multiple strategies on the data from October 2016 at the South tower. Our primary metric for determination of performance of these calibration strategies was the bias of the target tank (difference from the known value, averaged over the month). We also considered the standard deviation of the difference from the known value for the calibration cycles over the month. Through these "experiments", we address the following questions:

> 1) is the laboratory calibration for isotopic ratio (as described in Section 3.2) sufficient for our target compatibility level?

> 2) Does a field calibration of the mole fraction correction improve results?

> 3) Does using both the high tank and the low tank in the linear field calibration of the isotopic ratio improve results, or is it preferable to instead use the linear calibration determined in the laboratory, given the field calibration tanks used?

> 4) Do the analyzers drift significantly over the period of several hours, requiring multiple calibration cycles within each day?

After using the results of the field experiments to focus on a general calibration strategy, we further analyzed the results by testing the strategy, with variations, on all four towers, and evaluated the results again using the bias of the target tank as our primary metric. Although the low tank was planned to be used in the isotopic ratio calibration and the target as an independent test, we compare this strategy to that of instead using the target tank in the calibration and keeping the low tank as the independent test, given the isotopic ratios and sampling times utilized in the field for the majority of the deployment. In this case, the low tank was a metric for determining the compatibility of the isotopic ratio calibration.

**4.5 Cross-interference from other species**

**4.5.1 Overview**

The effects of cross-interference from other species must be considered for spectroscopic measurements. Rella et al. (2015) give proportional relationships for cross-interference from various species for the G2132-i analyzers. Listed in Table 2 are species with potential to affect the isotopic methane calibration, and their estimated effects for tower-based applications. We based these estimates on typical maximum values determined by flask (level at which 99 % of flask measurements at the South and East towers were below; for carbon monoxide, propane, butane, ethylene, and ethane), by in-situ measurements at the towers in this deployment (for water vapor and carbon dioxide), and by typical values (Warneck and Williams, 2012; for ammonia and hydrogen sulfide). There are no known ambient





estimates for methyl mercaptan (Barnes, 2015), so the odor threshold (Devos et al., 1990) was used as a maximum
value.
For the Picarro G-2132i analyzers, ethane contributed the largest interference and a correction to the isotopic ratio
was applied (Section 4.5.2). Because of water vapor effects, the sample was dried and the calibration gases
humidified. The effects of other species were neglected.
**4.5.2 Ethane correction**
Ethane ($C_2H_6$) is co-emitted with methane during natural gas extraction and its cross-interference with the isotopic
ratio of methane is significant. The magnitude of the effect of ethane on the isotopic methane is proportional to its
mole fraction and inversely proportional to the methane mole fraction. The two Scott-Marrin field calibration tanks
at each site were scrubbed of alkanes (including ethane), but the one NOAA/INSTAAR field calibration tank at each
site contained ambient levels of these species. Typical mole fractions of $C_2H_6$ (1.3 ppb) compared to the Scott-
Marrin tanks containing no ethane would lead to a 0.04‰ bias, if uncorrected. Furthermore, flask measurements at
the South and East towers indicated ethane up to 8 ppb, which corresponds to a 0.23 ‰ error.
The G2132-i analyzers reported an ethane measurement, but were not designed for high-compatibility C2H6
measurements at levels near background. In this deployment, 99 % of the flask measurements, which were taken in
the afternoon, were less than 8.0 ppb $C_2H_6$. In comparison, the drives near natural gas sources conducted by Rella et
al. (2015) indicated $C_2H_6$ mole fractions up to 13 ppm (note unit change). The ethane signal is subject to strong
cross-interference from water vapor, methane and carbon dioxide. Rella et al. (2015; Eq. (S20)) report coefficients
for these corrections. These coefficients indicate corrections larger in magnitude than the ethane mole fractions
measured in this deployment. We have thus not attempted to analyze the ethane results themselves. The ethane
output was however used to correct the isotopic methane data. To do so, we first developed a linear calibration using
the Scott-Marrin high field calibration tank containing zero ethane and the NOAA/INSTAAR target tank which we
assumed contained a background level of 1.5 ppb ethane (Peischl et al., 2016). This calibration is clearly a rough
estimate. Note that we determined the linear relationship between the reported ethane of each analyzer and its
calibrated value initially, and assumed that this relationship does not change throughout the deployment. Newer
models of the $\delta^{13}CH_4$ analyzer (G2210-i, Picarro Inc.) measure $C_2H_6$ at ppb levels, simplifying this correction
process.
We then corrected the isotopic methane for the effects of ethane cross-interference. For example, 1.3 ppb of ethane
in an air sample of 2 ppm $CH_4$ would, if uncorrected, shift the $\delta^{13}CH_4$ measurement higher by [+58.56 ‰ ppm
$CH_4$(ppm $C_2H_6$)$^{-1}$ x [0.0013 ppm $C_2H_6$]/[2 ppm $CH_4$]=+0.04 ‰. Note that the calibration coefficient for ethane has
been updated from that indicated in Rella et al. (2015). The correction to compensate for this error was applied to
all data, using the estimated ethane and measured methane values.



### 4.5.3 Water vapor and carbon dioxide

Water vapor can have a significant effect on the measurements of isotopic methane (up to ± 1 ‰ for up to 2.5 % $H_2O$) (Rella et al., 2015). Thus, the sample air was dried (<0.06 %) and the calibration gases slightly humidified (0.02 %) such that this effect is minimized (estimated to be < 0.02 ‰). For the range of ambient $CO_2$ observed in this study (~375 – 475 ppm), the difference from the calibration gases was ~100 ppm, and the effect was estimated to be < 0.03 ‰ (Table 2). The isotopic ratio of methane was thus not corrected for $CO_2$ effects.

### 4.5.4 Oxygen, argon, and carbon monoxide

The ambient variability in oxygen, argon, and carbon monoxide is expected to have a negligible effect on the isotopic ratio measurements (Rella et al., 2015) and no corrections for these constituents were applied to the isotopic methane data.

### 4.5.5 Other species

Ammonia, hydrogen sulfide, methyl mercaptan, propane, butane, ethylene are components of natural gas, but their cross-interference effects were small for our tower-based application for which the sources are relatively far from the measurement location. The effects of these species may be significant for other applications, such as automobile-based measurements. Like for ethane, the magnitude of the effect of these gases on the isotopic methane is proportional to the mole fraction of the contaminant species and inversely proportional to the methane mole fraction. In Table 2, maximum mole fractions from the flasks if available, or typical mole fractions from the literature, were used to estimate the effect of these species for our application. The cross-interference from these species was insignificant for our application, < 0.01 ‰.

### 4.6 Methods for determining enhancements

A background value is required to calculate enhancements in $CH_4$ and $\delta^{13}CH_4$. For this simple analysis, we chose a single tower to represent the background for the entire period. The predominant wind direction for the Marcellus region is from the west (Fig. 4). For westerly winds, the South tower is a reasonable choice for a background tower. The South tower measured the lowest overall mean afternoon methane mole fraction (1960.2 ppb $CH_4$). The mean afternoon methane mole fractions of the other towers, averaged only when data for the South tower exist, were 8.7, 7.0, and 2.9 ppb higher, at the North, Central, and East towers, respectively. As noted by Barkley et al. (2017), the area encompassing southwestern Pennsylvania and northeastern West Virginia contains large sources of $CH_4$, with emissions from conventional gas, unconventional gas, and coal mines all having significant contributions to the total. These large sources complicated the interpretation of the signals, as does changing wind direction. For this overview analysis, we calculated enhancements above the South background tower to determine overall signal



strength to compare with our target compatibility. The Central tower measured only mole fractions for the period
June – December 2016, so we focused on the period January – May 2016, when all sites measured both $CH_4$ and
$\delta^{13}CH_4$. We calculated probability distribution functions of the isotopic ratio and methane mole fraction
enhancements above the South tower in Section 6.8. We first examined the afternoon (defined here are 1700 – 2059
UTC), when the atmospheric is well mixed, allowing simpler interpretation of the measurements and more tractable
modeling. We then consider non-afternoon hours, when the atmosphere is less mixed and signals are typically
larger.
Keeling plots (Keeling 1961; Röckmann et al., 2016) are often used to infer the isotopic ratio of the methane source
as the intercept of the best fit line of the isotopic ratio as a function of the inverse methane mole fraction. In Section
6.8 we used this approach to estimate the source isotopic ratio of peaks observed during non-afternoon hours at the
Central tower.
**5 Methods for evaluating compatibility of in-situ tower measurements**
**5.1 Independent low tank**
While the low tank was planned to be used in the calibration of the isotopic ratio, the optimized calibration scheme
given the deployed tanks instead utilized the target tank in the calibration and kept the low tank as independent
(Section 6.3). The low tank was thus treated as an ambient sample. To evaluate the noise in the calibrated ambient
samples that results from noise in the calibration, we calculated the standard deviation over the period September 1 –
December 2 of the individual calibration cycles (6 min each), of the calibration cycles averaged over 1 day (81 min
total), and of the calibration cycles averaged over 3 days (4.1 hours total). These results are a proxy for the noise in
the calibrated ambient samples over those sampling periods. The same calculation was performed for the period
December 3 – December 31, a period during which an improved calibration tank sampling scheme was utilized.
**5.2 Round-robin testing**
Post-deployment round-robin style tests were completed in the laboratory in March 2017 for the analyzers
previously deployed at the North, Central and East Towers, in order to assess the compatibility achievable via our
calibration method. The analyzer deployed at the South tower was not included in these tests, as it was still in the
field. Two NOAA/INSTAAR tanks (JB03428: –46.82 ‰ $\delta^{13}CH_4$, 1895.3 ppb $CH_4$ and 381.63 ppm $CO_2$; and
JB03412: –45.29 ‰ $\delta^{13}CH_4$, 2385.2 ppb $CH_4$ and 432.71 ppm $CO_2$) were sampled for 70 min, with 8 min ignored
after each transition, and treated as unknowns. Additionally, high, low, and target tanks were sampled, with the
calibration applied as in the field for ambient samples (as described in Section 6.3). The high mole fraction tank
was sampled for 20 min and the low and target mole fraction tanks were sampled for 70 min, with 8 min ignored



after each gas transition. Four to six tests were completed for each analyzer. We used these tests as a means of
evaluating the compatibility of the analyzers, in terms of both mole fractions and the isotopic ratio.
**5.3 Side-by-side testing**
The precision and drift characteristics are not optimized for $CO_2$ for the G2132-i analyzers, compared to the G2301
analyzers, which measure $CO_2$, $CH_4$ and $H_2O$ mole fractions. To test the performance of the G2132-i analyzers for
consideration of the data for use as part of the continental-scale $CO_2$ network, G2301 and G2132-i (Picarro, Inc.)
analyzers were run side-by-side for one month (June 2016) at the South tower. The sampling system for the G2132-
i was as described in Section 4.3. A separate ¼" (0.64 cm) tubing was used for the G2301 analyzer and an intercept
calibration using the target tank is applied daily. The sample air for the G2301 analyzer was not dried and the on-
board water vapor correction was used. This testing was used to evaluate the mole fraction compatibility,
particularly for $CO_2$, of the G2132-i analyzers compared to the G2301 analyzers.
**5.4 Flask measurements**
Flask measurements were used for independent validation and error estimation of the continuous $CO_2$, $CH_4$ and
$\delta^{13}CH_4$ in-situ measurements. In addition, the flasks were measured for a suite of species including $N_2O$, $SF_6$, CO,
$H_2$ (Conway et al., 2011), halo- and hydro-carbons (Montzka et al., 1993) and stable isotopes of $CH_4$ (Vaughn et al.,
2004). The flasks were filled over a 1-hour time period in the late afternoon (1400–1500 LST), thereby yielding a
more representative measurement compared to most flask sampling systems, which collect nearly instantaneous
samples (e.g., ~10 sec). Samples were measured only when winds were blowing steadily out of the west or north
(~45–225°) to ensure that the samples are sensitive to and representative of the broader Marcellus shale gas
production region that is the focus of this study.
**6 Results**
**6.1 Allan standard deviation results**
As described in Section 3.1, two tanks were sampled for 24 hours each to determine the Allan standard deviation as
a function of averaging interval for the G2132-i analyzers. The resulting Allan standard deviations for $\delta^{13}CH_4$, $CH_4$
and $CO_2$ are shown in Fig. 6. For the high tank, the Allan deviation for $\delta^{13}CH_4$ (Fig. 6A) was < 0.2 ‰ (our target
compatibility) for an averaging interval of 2 min (the averaging interval used each field calibration cycle of the high
tank). To reduce the noise to < 0.1 ‰, an averaging interval of 4 min is sufficient (in addition to the time required
for the transition between gases). For the low tank, in order for the Allan standard deviation to be < 0.2 ‰, 32 min
were required and 64 min for 0.1 ‰ noise.



For $CH_4$ (Fig. 6B), both the high and low tank Allan deviation was < 1 ppb for even a 1-min averaging interval. The
$CO_2$ levels in the high and low tanks were similar (~400 ppm), and an averaging interval of 6 min corresponded to
Allan standard deviations of 0.3 ppm, and 64 min were necessary for 0.1 ppm (Fig. 6C).
While in some cases, analyzer drift may contribute to increased Allan standard deviations as the averaging interval
increases, we do not see evidence of this for the averaging intervals shown.
**6.2 Errors prior to laboratory calibration**
The methane isotopic ratio was calibrated in the laboratory prior to deployment, as described is Sections 3.2.1 and
3.2.2. Averaged methane isotopic ratios prior to these calibrations are shown in Fig. 7. An error in isotopic ratio as a
function of isotopic ratio is apparent, as is the mole fraction dependence of the isotopic ratio response.
**6.3 Optimized calibration scheme determination**
As described in Section 4.4, several calibration strategy "experiments" were conducted using the data from October
2016 for the South tower to determine the optimized procedure for utilizing the field calibration data for isotopic
methane. The results from these experiments (EXPTs) are shown in Table 3. In EXPTs A–G, the daily average of
the 13.5 calibration cycles per day for the high tank and low tank was used (if applicable), and the target was
independent of the calibration. Of the EXPTs A–G, EXPT D (laboratory calibration, and high and low tanks for
mole fraction correction) gave the results with the lowest bias and standard deviations of the difference from the
known value: –0.3±0.4 ‰ for the target tank. The standard error for EXPT D was 0.1 ‰. Therefore, the mean was
significantly improved in EXPT D, compared to the EXPTs A–C and G, with biases ranging from –2.2 to –0.5 ‰.
EXPTs E and F showed comparable bias, but larger standard deviations.
With only the factory calibration (EXPT A), the bias of the values averaged over the month was –0.6 ‰ ±1.3 ‰ for
the target tank. Here the standard deviation was calculated for all of the calibration cycles over the month, e.g., 67
% of the calibrations cycles yielded differences from known values between –1.9 and 0.7 ‰ for the target tank using
EXPT A. Without applying any slope calibration either in the laboratory or in the field, but applying a mole fraction
correction (EXPT B), the standard deviation was improved to 0.4 ‰, but the bias for the target tank was increased to
–2.2 ‰. With a linear calibration in the field, but no laboratory or mole fraction calibration (EXPT C), the mean
bias for the target tank was improved to –0.5 ‰, but the standard deviation of the calibration cycles was high (1.4
‰).
EXPT E used only the laboratory calibration (completed prior to deployment of the analyzers). The bias for the
target tank was the same as for EXPT D, but the variation of the errors was larger, with a standard deviation of 1.2



‰. We thus answer question 1) in Section 4.4: field calibration of the isotopic ratio significantly improved the
variation in the tank differences from the known value over the month.
In EXPT F, neither the low nor the target tank were used in the field calibration of the isotopic ratio, and a field
calibration of the mole fraction correction was not applied. The bias and standard deviation of the calibration cycles
for the target tank were larger (–0.4±0.9 ‰) than for EXPT D, indicating that field calibration of the mole fraction
correction (and thus having both high and low field calibration tanks) was beneficial (question 2 in Section 4.4).
EXPT G, with no laboratory calibration, using both the high and low tanks to apply a linear calibration, and to apply
the mole fraction correction yielded results that indicated larger bias and standard deviation in the target tank (–
0.5±1.4 ‰) compared to EXPT D. Thus, using the low tank in the application of the linear calibration was not
beneficial in this case compared to using the laboratory calibration (question 3 in Section 4.4). It did, however,
significantly improve results compared to the case with no laboratory calibration of the linear term (i.e., EXPTs A
and B). Following EXPT G but using two, instead of one, high mole fraction tanks (with different isotopic ratios)
for the linear calibration would likely yield similar results to EXPT D, and would require an additional field
calibration tank (~10 ppm and –23.9 ‰, for example), but has the advantage of not requiring a laboratory
calibration. We note that in the present case, we used two measurements to solve for three unknowns: $p_1$, $c_0$ and $\chi$.
They were thus not independent.
In EXPT D1, only the first high and low tank calibration cycle of the day was used, rather than the average of the
13.5 daily calibration cycles. The results, in terms of the one-month mean bias and standard deviation were similar
to EXPT D, using the mean of all 13.5 daily calibration cycles. In EXPT D-T, only the high and low tank
calibration cycle immediately preceding the target tank sampling was used. Again, the results were similar to EXPT
D, indicating that the mean results over the month are not significantly affected by using only one calibration cycle
per day for the field calibration, thus answering question 4 in Section 4.4 – the analyzers did not appear to drift
significantly over several hours and thus multiple calibration cycles within each day are not required.
We now expand the analysis of the best performing calibration scheme (EXPT D and variations) to the other towers
(Table 4). First, we show the results for EXPT H, with only the laboratory calibration prior to deployment, for
comparison. The bias of the differences from known values varied amongst the towers, with magnitudes ranging
from 0.0 to 1.9 ‰. For EXPT D, the bias of the one-month data of the target tank varied between –0.3 and –0.8 ‰
among the towers. The target tank contained typically ambient levels of isotopic methane, and we want to minimize
biases in this range.
Since we found that the difference between EXPT D and its variants was not significant, only one calibration cycle
per day was necessary (although longer sampling would be preferable based on Allan deviation results). So instead
of using the low tank for the calibration and keeping the target tank independent, we instead used the target tank for
the calibration and keep the low tank independent. EXPT D-HI-PRE-HI-T was similar to EXPT D, but used only





the high tank value immediately preceding the target tank sampling for the day and used the target tank for the mole
fraction correction.  The results indicate that the center of the maximum bias was shifted from the target tank (about
–47.2 ‰) to the low tank (about –23.9 ‰).  This method is preferable since the ambient sample is near the target
tank values.  Using the daily average for the high instead of just one calibration cycle (EXPT D-HI-T) indicates
similar results.  We thus chose to use the EXPT-D-HI-T protocol, but sampling the high and low tanks every 90
minutes does not appear to be necessary – using the daily average is sufficient, i.e., the analyzers do not appear to
drift significantly in the period of one day.
As an example, the tank results (differences from known values) using only the laboratory calibration for isotopic
ratio and following the EXPT-D-HI-T protocol are shown in Fig. 8 for the period September – December 2016.  For
the results using only the laboratory calibration, analyzer drift is apparent for all three tanks.  Without a field
calibration, the isotopic ratio was biased by up to 2 ‰.  The target tank was sampled only once per day and the
resulting measurement is used in the calibration; hence the apparent drift following final calibration is necessarily
zero.  On the other hand, the high tank was sampled 13.5 times per day and the average is used in the calibration.
The low tank was independent of the isotopic ratio calibration.  Prior to 3 December 2016, the low tank was also
sampled 13.5 times per day, for 6 min each time (excluding transition time between gases).  Thus, the noise apparent
in Fig. 8B prior to that date is at least partially due to insufficient sampling times.
On 3 December 2016, an improved sampling strategy was implemented, in which the target tank sampling time was
increased from 6 min/day to 54 min/day (excluding transition times), achieved by sampling for 20 min every 420-
min cycle (3.4 times/day, on average).  The calibration times were achieved with multiple cycles in order to avoid
not sampling the atmosphere for long periods.  The calibration data for each day are averaged and applied to the
ambient data.  The low tank was sampled using an identical strategy (20 min every 420-min cycle), with the total
amount of sampling time per day changing from 81 min to 54 min.  The high tank was sampled on average 1.7 times
per day (every 840 min) for 10 min.   Excluding the transition times, the high tank sampling time was thus reduced
from 26 min/day to about 10 min/day.  This strategy reduced the noise apparent in Fig. 8B, but is not expected to
affect the long-term (e.g., scales of several days or longer) bias.
The relative effects of the calibration terms are illustrated in Fig. 9.  The terms $c_0$ (Fig. 9B) and $\chi$ (Fig. 9C) in Eq.
(3) are time-dependent drift terms.  These terms vary because of spectral variations in the optical loss of the empty
cavity ($c_0$), and because of errors in the temperature or pressure of the gas, or changes in the wavelength calibration
($\chi$).  Recall that the parameters $c_0$ and $\chi$ were calculated following Eq. (15) in Rella et al. (2015).  The calculation of
the parameter $c_0$ used measurements from the high and target tank.  The calculation of the parameter $\chi$ used
measurements of the high tank and was not independent from $p_0$.  The largest calibration effect was from the $c_0$
term, which increased the calibrated isotopic ratios by –0.5 to 4 ‰ during September to December 2016.  The $\chi$
term increased the final calibrated isotopic ratios by a smaller amount, –0.6 to 0.2 ‰.  There is variability in the
calibration effect of these terms, although no software or hardware changes were applied during this period.





## 6.4 Noise in independent low tank as a function of averaging interval

In the optimized calibration scheme, the low tank was independent, and was treated as an ambient sample. The low tank differences from known values, averaged over differing intervals, are shown in Fig. 10. As described in Section 5.1, the standard deviations of the low tank differences are a proxy for the noise in the calibrated ambient samples over those averaging intervals. The standard deviation of 13.5 calibration cycles per day, each of 6 min length, over the period September 1 – December 2 is 0.62 ‰. During this period, the calibration used 6 min/day measurements of the target tank. The standard deviation of the low tank calibration cycles was similar to expectations based on the Allan standard deviation (Fig. 6). Averaging over the cycles in one day (a total of 81 min of data) yielded a standard deviation of 0.40 ‰. Based on this result, differences in the hourly average between towers of less than 0.40 ‰ were likely not significant. For 3-day means (a total of 4.1 hr), the standard deviation over the three-month period was 0.26 ‰. For the period after the calibration tank sampling scheme was improved (primarily by sampling the target tank for 54 min/day instead of 6 min/day), December 3 – December 31, the standard deviation of the individual cycles reduced substantially, to 0.25 ‰, and that of the one-day (three-day) mean of the cycles was 0.18 ‰ (0.11 ‰). Therefore, according to this metric, after the improved calibration scheme was implemented, differences in the hourly average between towers of greater than 0.18 ‰ were significant.

## 6.5 Round-robin testing

The results for the round-robin style laboratory testing of two NOAA/INSTAAR tanks are shown in Fig. 11. The mean of the errors (measured – NOAA known value) calculated from the results of four to six tests for each analyzer were –0.08 to 0.04 ppm $CO_2$ within the 0.1 ppm WMO compatibility recommendation for global studies of $CO_2$ (GAW Report No. 229, 2016). The standard error, indicating an estimate of how far the sample mean is likely to be from the true mean, for the means of the $CO_2$ tests were 0.03 – 0.10 ppm. The mean difference was –0.03 to 0.02 ppm $CO_2$ for the analyzers, averaged over the two round-robin tanks (analogous to averaging over the entire range of $CO_2$ during the flask comparison, for example). For $CH_4$, the means of the errors were 0.03 – 0.07 ppb $CH_4$, for the NOAA/INSTAAR tank measuring 2385.2 ppb, and –0.83 to – 0.70 ppb $CH_4$ for the NOAA/INSTAAR tank measuring 1895.3 ppb $CH_4$. Therefore, there was a slight error in the slope of the linear calibration, possibly attributable to tank assignment errors. However, the error was well within the WMO recommendations for global studies of 2 ppb $CH_4$ (GAW Report No. 229, 2016), and the range of NOAA/INSTAAR tanks encompassed the majority of the $CH_4$ mole fraction observed during the study. We also note that the standard error for the means of the $CH_4$ tests were 0.07 – 0.12 ppb. Averaging over the two round-robin tanks, the mean difference was –0.40 to – 0.32 ppm $CH_4$ for the analyzers. For $\delta^{13}CH_4$, the mean errors for each analyzer/tank pair were –0.33 to 0.24 ‰ for these tanks within the range of ambient isotopic ratio and the standard errors were 0.05 – 0.10 ‰. The mean errors were –0.14 to 0.03 ‰ for each analyzer.



## 6.6 Side-by-side testing

Side-by-side testing of a G2301 ($CO_2$/$CH_4$/$H_2O$) analyzer and a G2132-i analyzer ($CH_4$/$\delta^{13}CH_4$/$CO_2$) for June 2016 at the South tower resulted in mean differences of 0.06±0.41 ppm $CO_2$ and 0.9±1.5 ppb $CH_4$, with the G2132-i analyzer measuring slightly lower for both species. Here the standard deviation was based on the 10-min average calibrated values for the month for all times of the day. The standard error of the differences was 0.01 ppm $CO_2$ and 0.02 ppb $CH_4$. These results indicate that the performance of the G2132-i is similar for $CO_2$ and $CH_4$ mole fractions, at least in terms of the long-term mean. In terms of utilizing the mole fraction data in atmospheric inversions, the multi-day mean afternoon differences are more appropriate. The five-day mean afternoon difference for the month was 0.05±0.08 ppm $CO_2$ and –0.7±0.1 ppb $CH_4$. The G2132-i analyzers are thus appropriate for use in the atmospheric inversions and in the global network where 0.1 ppm $CO_2$ and 2.0 ppb $CH_4$ have been identified as criteria. For these results, recall that the target tank was sampled for a total of 30 min in five days. To optimize results on a daily time scale, sampling the target tank for 60 min per day would be preferable for improving $CO_2$ results. We also note that round robin testing of these instruments requires 60 min sampling per tank.

## 6.7 Flask to in-situ comparison

For January – December 2016, the mean flask to in-situ $CH_4$ difference at the East tower was –1.2 ± 2.2 ppb $CH_4$, and at the South tower was –0.9 ± 1.4 ppb $CH_4$ (Fig. 12A). Recall from Section 5.4 that flasks were sampled in the late afternoon, integrated over one hour, and only when the winds were steadily from the west or north. Here the standard deviation reported is that of the hourly flask to in-situ differences for the year. Thus, at the South tower, for example, on 67% of the sampled afternoons indicated differences for $CH_4$ within 1.4 ppb of the mean of –0.9 ppb. For $CH_4$, data points with high temporal variability (standard deviation of raw ~2 sec data within the hour > 20 ppb) were excluded, on the basis that the ambient variability was large, making comparisons difficult. The standard error was 0.24 ppb at the East tower and 0.14 ppb at the South tower. Thus, there is high confidence that the difference between the in-situ and flask measurements at both towers is more compatible than the WMO recommendation. As for the side-by-side testing, the G2132-i analyzers were slightly lower than the "known", in this case, the flask results. The difference, was however, less than the target compatibility, and the flasks could in theory be biased.

Although $CO_2$ is not the focus of this paper, the differences were –0.21 ± 0.31 ppm for the East tower and 0.21 ±0.35 ppm for the South tower (Fig. 12B). The standard error was 0.03 ppm at the East tower and 0.04 ppm at the South tower. The magnitude of $CO_2$ differences was somewhat larger in the growing season. The mean flask to in-situ differences are thus larger than the WMO recommendation of 0.1 ppm, but at the extended compatibility goal of 0.2 ppm $CO_2$ (GAW Report No. 229, 2016).



For the isotopic ratio of methane, the mean flask to in-situ differences were $0.08 \pm 0.54$‰ and $0.02 \pm 0.38$‰ at the
East and South towers, respectively (Fig. 12C). The standard error was 0.06‰ and 0.04‰ at the East and South
towers, respectively. Thus, there is high confidence that these differences are less than the target compatibility of
0.2 ‰. The standard deviation reported is that of the hourly flask to in-situ differences for the year. The range of
$\delta^{13}CH_4$ throughout the project (including day and night) was relatively small: one standard deviation (67%) of the
data points are between 46.7 – 48.2 ‰, a range of 1.5 ‰. Errors for isotopic ratios outside the calibration range
(further from the high and target calibration tanks) would likely be larger. For example, the mean error of the
independent low tanks (averaging over all calibration cycles during a one month period) at the towers (Table 4) were
0.2 – 0.7 ‰.
The flask ethane results (not shown) indicate that 90% of the afternoon values are < 4.5 ppb $C_2H_6$. The G2132-i
analyzers were not optimized for these relatively low enhancements of ethane, the ethane calibration for the in-situ
analyzers as described in Section 4.5.2 was designed only for applying a correction to the isotopic methane data, and
the flask to in-situ comparison was poor.
**6.8 Observed enhancements**
We now examine the overall enhancements in methane mole fraction and isotopic ratio at the towers. In the first set
of plots, we focus on the majority of the afternoon data points by truncating the scale for the probability distribution
functions of methane mole fraction and isotopic ratio (Fig. 13A, B, D, E, G, and H). The time scale of the individual
data points was 10 min and the data were afternoon only (1700–2059 UTC, 1200–1559 LST) for the time period
January – May 2016, a period during which all four towers measured both methane and isotopic ratio. The median
enhancements for both isotopic ratio (–0.15 to 0.12 ‰) and methane mole fraction (less than 1 ppb) were less in
magnitude than the compatibility of the analyzers. This result is consistent with the results of Barkley et al. (2017),
who found the emission rate of methane due to natural gas extraction activities to be very low, 0.36 % of total
production. The standard deviation of 10-min segments of isotopic ratio enhancements was 0.8 ‰ at each of the
towers. We note that the Allan standard deviation for 10-min averaging times for ambient levels of methane was 0.4
‰ $\delta^{13}CH_4$. For the daily afternoon averages, the standard deviation of the daily afternoon averages was reduced
only slightly, to 0.6 – 0.7 ‰. Thus the majority of the observed width of the distribution appears to be real. For
isotopic ratio, 43 – 54 % of the 10-min segments were greater than 0.6 ‰ in magnitude (3 times the target
compatibility) (Fig. 13A, D, and G) and are thus detectable by the analyzers. The standard deviations of the
methane mole fraction enhancements were 60.7, 30.0 and 33.8 ppb for the North, Central, and East towers,
respectively (Fig. 13 B, E, and H). 57 – 66 % of the data points indicated enhancements greater than 6 ppb $CH_4$ in
magnitude (3 times the target compatibility of 0.2 ‰) for the North, Central, and East towers, respectively (Fig. 13
B, E, and H) and are thus detectable. The majority of afternoon data points indicated relatively few local sources of
contamination.



There are however a few outliers during the time period with large values above the background tower (up to 1500 ppb enhancement at the North tower). The isotopic as a function of inverse methane mole fraction at each non-background tower are shown in Fig 13C, F, and I.

During non-afternoon hours (0000–1659 and 2100–2359 UTC), the median isotopic ratio enhancements above the South tower were still indistinguishable from zero (Fig. 14A, D, and G). The median methane mole fraction enhancement was slightly higher than during the afternoons, at 3.5, 6.8, and 9.8 ppb for the North, Central, and East towers, respectively (Fig. 14B, E, and H). There were however more outliers, particularly at the Central tower (Fig. 14C, F, and I). Applying a best fit line to all of the data shown in Fig 14F gave a poor correlation coefficient ($r^2$=0.22) because there were many data points with no local sources.

The isotopic ratio as a function of inverse methane mole fraction (i.e., the Keeling plot) for each of the eight largest peaks in the non-afternoon methane data are shown in Fig. 15. The intercepts of the best fit lines for the peaks indicate that the sources contributing to the peaks gave a mean isotopic ratio of –31.2 ‰, with values ranging from –33.2 to –28.0 ‰. The correlation coefficients were high ($r^2$=0.92 – 1.0), except for one peak, which was excluded from the mean.

The footprints of towers are large compared to mobile measurements near the ground, for example, which is ideal for determining regional emissions. Thus, specific emissions sources with specific isotopic signatures are diluted by mixing, making the enhancements above background very small. For these eight peaks, the peak heights were 334.1 – 2007.8 ppb $CH_4$ and 2.5 – 8.7 ‰. The reduced methane enhancement at other data points within the peak was indicative of increased dilution of the local source by the background signal.

**7 Discussion and Summary**

We found that field calibrations (including both a linear calibration and a correction for mole fraction dependence of $\delta^{13}CH_4$) significantly improved the compatibility of the measurements (as seen by comparing EXPTs H and F with EXPT D in Table 3). There was, however, no significant drift within a single day (as seen by comparing variations of EXPT D in Table 4). Using these findings, we developed an optimized calibration strategy, given the tank sampling strategy used in the deployment prior to 3 December 2016 (Table 5). Instead of using the target tank as an independent assessment of compatibility, we used the target tank in the calibration scheme and evaluated performance with the low tank at each site. An improved tank sampling strategy was implemented on 3 December 2016.

Prior to the improvement in the tank sampling strategy, averaging over a period of 4.1 hours within a 3-day time period (a proxy for the noise within 4-hour afternoon averages of ambient data) yielded a standard deviation of the





independent low tank of 0.26 ‰. After the improvements in the tank sampling strategy were implemented,
averaging over the same time period was sufficient to achieve standard deviation of 0.11 ‰.
Furthermore, an ethane rough calibration was performed using the two field tanks scrubbed of ethane and one field
tank with ambient levels of ethane. These roughly calibrated ethane values were subsequently used to correct for
cross-interference with the isotopic ratio of methane.
The round robin results using NOAA/INSTAAR tanks treated as unknowns in Fig. 11 showed mean error (averaged
over the two round robin tanks) of –0.03 to 0.02 ppm $CO_2$ and –0.40 to –0.32 ppb $CH_4$ for the analyzers. These
results were within the WMO recommendations (GAW Report No. 229, 2016). For $\delta^{13}CH_4$, the mean errors of the
tests for each analyzer were –0.14 to 0.03 ‰, with larger errors for individual tanks. Earlier round-robin style
testing with reduced sampling times of about 10 min per tank (not shown) indicated increased errors for $CO_2$ and
$\delta^{13}CH_4$, consistent with the Allan standard deviation results (Fig. 6).
The flask to in-situ comparison for $\delta^{13}CH_4$ (Fig. 12) showed that the field calibration tank sampling strategy and
calibration protocol (Table 4) used in this study was sufficient for producing isotopic methane results with low long-
term bias (0.02 to 0.08 ‰ averaged over one year). The $CO_2$ differences were –0.21 to 0.21 ppm and the $CH_4$
differences were –1.22 to –0.87 ppb.
Recall from Table 1 that the calibration of the G-2132i analyzers require consideration of 1) the isotopic ratio linear
calibration (preferably using high mole fractions tanks), 2) the mole fraction dependence of the isotopic ratio
calibration (using high and low mole fraction tanks), 3) the correction due to ethane cross interference (using one
tank without ethane and one tank with ambient ethane), and 4) drift in the $CO_2$ and $CH_4$ mole fractions (using at
least one tank near ambient isotopic ratio and mole fraction). The isotopic ratios and $CH_4$ mole fractions of these
tanks as used in the present deployment are graphically represented in Fig. 16A. Prior to implementation of the
improving field calibration tank sampling strategy, the high and low tanks were sampled for 26 and 52 min/day
(excluding transition time between gases), as listed in Table 5. The sampling times for the high and low standards
are sufficient for Allan deviation < 0.1‰, but neither of those tanks were at ambient ranges of $\delta^{13}CH_4$. If the
calibrations and analyzer response were both linear, we would expect about 0.1 ‰ error in the target tank if kept
independent, but EXPT D in Table 4 indicate a bias between –0.3 and –0.8 ‰, which is very large compared to the
ambient enhancements observed (one standard deviation of the tower measurements at all times of day were
between –48.2 and –46.7 ‰). Thus, we instead chose to minimize mean error at ambient values (target tank) rather
than at the isotopic ratios of the low tanks (–23.9 ‰). This procedure added noise to the ambient data because the
daily sampling time for the target tank was only 6 min/day. Note that the high mole fraction tanks are ideally near
ambient isotopic ratio, but the effect of being further from ambient isotopic ratios is likely small. The isotopic ratios
of the high tanks are thus listed as values provided by Isometric Instruments bottles, making the tanks logistically
less challenging and less expensive to acquire. On 3 December 2016, we implemented an improved sampling





strategy, primarily by increasing the sampling time for the target tank (Table 5). An alternate possible calibration
tank sampling strategy is to sample an additional high tank at a different isotopic ratio (Table 5, third column, and
Fig. 16B). With this strategy, a laboratory calibration prior to deployment is not necessary. Also, both the slope
and intercept of the linear calibration can be adjusted in field, rather than just the intercept, which may improve
results. In this case, however, the calibration is over-constrained, using two tanks to solve for two variables.
Additionally, it would also be preferable for both low and target tanks to be near –47 ‰ and sampled for about one
hour per day, either all at once or spread out over the day. The ranges of isotopic ratios and $CH_4$ mole fractions
desirable are indicated in Fig. 16B.
The requirements to achieve compatibility for $\delta^{13}CH_4$ for tower measurements as described in this paper are
demanding. Source attribution using mobile measurements, rather than tower measurements, is less demanding due
to the relatively large ambient signals typically encountered. If compatibility of 0.5 ‰ is sufficient for a specific
application, the sampling strategy listed in fourth column of Table 5 and Fig. 16C may be applicable. In this case,
the noise is increased by reducing sampling time requirements. Another possible simplification is to utilize the
isotopic ratio of the Isometric Inc. bottle used to spike the high and low tank, rather than calibrating them in the
laboratory. The calibration of the tanks is shown to be <0.4 ‰ magnitude different from the bottle value (Table 1),
and varies by tank. If, however, one analyzer is used to sample a plume, this bias is unimportant.
An alternative calibration approach is to separately calibrate the individual isotopologues (in this case, $^{13}CH_4$ and
$^{12}CH_4$ dry mole fractions) (Flores et al., 2017). This approach has the advantage of simple calibration equations, but
has the disadvantage that the quantities of interest (e.g., total mole fraction and isotopic ratio are calculated rather
than directly calibration.) Like the approach applied in this paper, it also requires at least two standard tanks, and
could utilize an independent tank for testing.
The signals observed in the study region were generally small, but the isotopic ratio enhancements were larger than
would be expected based on the methane mole fraction enhancements from local sources. For afternoon hours at the
Central tower, for example, 43 % of the enhancements in $\delta^{13}CH_4$ were detectable above background with
magnitudes > 0.6 ‰, 3 times the analyzer compatibility. For a thermogenic source with isotopic ratio of –35 ‰, a
background isotopic ratio of –47 ‰, and assuming a measured $CH_4$ mole fraction of 2000 ppb, a measured isotopic
ratio enhancement of –0.6 ‰ corresponds to a 100 ppb peak in $CH_4$ above background, following Eq. (1).
Enhancements in $CH_4$ of 100 ppb were rarely encountered, however (Fig. 13B, E, and H). Using Eq. (1) to predict
enhancements of isotopic ratio based on the observed methane mole fraction enhancements corresponded to only 3
% of the isotopic ratio enhancements expected to be > 0.6 ‰ in magnitude. Thus during the afternoon hours, the
deviations from background were not likely directly from local sources. We also note that we focused on the period
January – May 2016 in this work. Larger enhancements were observed in the latter half of 2016.



During the morning hours, however, several peaks resulting from local sources were observed.  The mean source
isotopic signal indicated by Keeling plot analysis of the eight largest peaks at the Central tower was –31.2 ‰, fairly
heavy even for oil/natural gas sources.  In general, the isotopic signature for natural gas sources varies from region
to region, and even within one region.  The mean isotopic ratio of methane in gas wells in the northeastern
Pennsylvania section of the Marcellus region was shown to vary by depth, from –43.42 ‰ with a standard deviation
of 6.84 ‰ for depths of 0 to 305 m, to –32.46 ‰ with a standard deviation of 3.84 ‰ for depths greater than 1524 m
(Baldassare et al., 2014).  Similarly, Molofsky et al. (2011) found that the isotopic signatures of gases from the
deeper layers of the Marcellus Shale in Susquehanna County, Pennsylvania, to be heavier than the shallower Middle
and Upper Devonian deposits, with values for the deep layers ranging from –30 to –21 ‰.  Thus, the source
signature determined here is consistent with a natural gas source originating from deep wells in the Marcellus
region.  The peaks occurred during the morning hours, when the boundary layer is typically stable, making modeling
more difficult, and the winds prior to the peaks were not from a consistent direction.  Determining the location of the
specific emitter(s) contributing to these peaks is thus beyond the scope of this paper.
In this paper, we present details of the laboratory and field calibrations of cavity ring-down spectroscopy (Picarro,
Inc, G2132-i) analyzers measuring $CH_4$, $\delta^{13}CH_4$, and $CO_2$.  Numerous calibration schemes were tested, and the
effects on measurement compatibility of each scheme were quantified.  We have demonstrated that the network of
four G2132-i analyzers, deployed on towers in the Marcellus Shale region in Pennsylvania, was able to detect the
small differences in $CH_4$ and $\delta^{13}CH_4$.
**Data Availability**
Miles, N.L., D.K. Martins, S.J. Richardson, T. Lauvaux, K.J. Davis, B.J. Haupt, and C. Rella, 2017. In-situ tower
atmospheric methane mole fraction and isotopic ratio of methane data, Marcellus Shale Gas Region, Pennsylvania,
USA, 2015-2016. Data set. Available on-line [http://datacommons.psu.edu] from The Pennsylvania State University
Data Commons, University Park, Pennsylvania, USA. http://dx.doi.org/10.18113/D3SG6N.
**Competing Interests**
TL, SJR, NLM, and KJD are co-owners of a related company, Carbon Now Cast, LLC.
*Acknowledgments.*  The authors thank B. Vaughn and S. Englund Michel (Institute of Arctic and Alpine Research,
University of Colorado) for providing analysis of methane isotopic ratios of the flask data and for advice regarding
gas handling techniques involving isotopic ratios.  The authors also acknowledge R.P. Barkley (Tunkhannock Area
Middle School) for his contributions to maintaining instrumentation at the tower sites.  This work was funded by the
Department of Energy National Energy Technology Laboratory (DE-FOA-0000894).





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





**Tables**
Table 1. Field calibration tanks used at the tower locations. The high and low tanks were planned to be used for the field
calibration of $\delta^{13}CH_4$, with the target as an independent test. However, as described in Section 6.3 and indicated in the
table, the high and target tanks were used for the field calibration of $\delta^{13}CH_4$. Only the target tank is used for field
adjustment of the $CH_4$ and $CO_2$ mole fraction calibration. The $CH_4$ and $CO_2$ mole fractions for the high and low tanks
are less certain than that of the target tanks.
*Determined via laboratory measurements.
**NOAA/INSTAAR calibration (WMO X2004A scale for $CH_4$ and WMO X2007 for $CO_2$).
*** Field calibration – values not used.

| Tank number | Deployment location | Measured isotopic ratio $\delta^{13}CH_4$ (‰) | $CH_4$ mole fraction (ppb) | $CO_2$ mole fraction (ppb) | Used for field calibration of $\delta^{13}CH_4$ | Independent test of $\delta^{13}CH_4$ calibration | Used for field adjustment of $CH_4$ and $CO_2$ mole fraction calibration (intercept only) | Used for ethane correction |
|---|---|---|---|---|---|---|---|---|
| CA06418 | North-High | -38.31* | 9701* | 397.75*** | ✓ | | | ✓ |
| CA05551 | North-Low | -23.67* | 1926.8* | 402.70*** | | ✓ | | |
| CB10825 | North-Target | -47.26** | 1867.59** | 399.71** | ✓ | | ✓ | ✓ |
| | | | | | | | | |
| CA05419 | Central-High | -38.48* | 10534* | 399.66*** | ✓ | | | ✓ |
| CA06438 | Central-Low | -23.80* | 2064.6* | 397.82*** | | ✓ | | |
| CB10734 | Central-Target | -47.25** | 1878.53** | 397.09** | ✓ | | ✓ | ✓ |
| | | | | | | | | |
| CA05330 | South-High | -38.68* | 10152* | 403.10*** | ✓ | | | ✓ |
| CC114999 | South-Low | -23.72* | 1999.2* | 402.58*** | | ✓ | | |
| CB10727 | South-Target | -47.24** | 1868.33** | 399.68** | ✓ | | ✓ | ✓ |
| | | | | | | | | |
| CA06410 | East-High | -38.52* | 10414* | 407.45*** | ✓ | | | ✓ |
| CA06357 | East-Low | -24.02* | 2079.7* | 368.47*** | | ✓ | | |
| CB10718 | East-Target | -47.26** | 1867.94** | 399.67** | ✓ | | ✓ | ✓ |



Table 2. Maximum error estimate attributable to cross-interference, based on typical values for this tower-based
application and estimated effects on CRDS measurements (Rella et al., 2015), and assuming 2 ppm ambient $CH_4$ mole
fraction. Typical maximum values determined by flask[f] (level at which 99 % of (afternoon) flask measurements at the
South and East towers are below), by in-situ measurements at Marcellus towers[i], or by typical values[t] (Warneck and
Williams, 2012). [a]No known ambient estimates (Barnes, 2015) / odor threshold (Devos et al., 1990).

| Gas Species | Typical maximum value or range | Estimated maximum error |
|---|---|---|
| Carbon monoxide | Range[f]: 107.5-200.7 ppb | 0.01‰ |
| Water vapor, dried sample | Range[i]: 0.02 – 0.06% | 0.02‰ |
| Water vapor, ambient moisture | Range: 0 – 2.5% | ±1‰ (Rella et al., 2015) |
| Carbon dioxide | Range[i]: 375 – 475 ppm | 0.03‰ |
| Propane | Max[f] 3.6 ppb | 0.01‰ |
| Butane (i-Butane + n-Butane) | Max[f] 1788 ppt | 0.01‰ |
| Ammonia | Typical[t] 90 ppt | 0.01‰ |
| Hydrogen sulfide | Typical[t] 30 ppt | 0.01‰ |
| Methyl mercaptan | Odor threshold[a]: 1 ppb | 0.01‰ |
| Ethylene | 13.0[f] ppt | 0.01‰ |
| Ethane | Max[f] 8.0 ppb (typical background[t]: 1.3 ppb) | 0.23‰ (0.04‰ typical) |



**Table 3.  Results for the South tower for October 2016 using multiple possible isotopic ratio calibration schemes.  In all of**
**these experiments, the target tank isotope value was independent of the analyzer calibration, as planned prior to the**
**deployment.  This differs from the optimized calibration scheme and tank uses listed in Table 1, as discussed in the text.**
**In EXPTs A – G, the daily average of the 13.5 calibration cycles per day for the high tank and low tank was used (if**
**applicable).  In EXPT D1, only the first high and low tank calibration cycle of the day was used.  In EXPT D-T, only the**
**high and low tank calibration cycle immediately preceding the target tank sampling was used.  The high tank at each**
**tower contains 9.7–10.5 ppm CH4 and about –38.3 ‰ $\delta^{13}$CH$_4$, the low tank contained 1.9–2.1 ppm CH$_4$ and about –23.9**
**‰ $\delta^{13}$CH$_4$, and the target contained ~1.8 ppm CH4 and about –47.2 ‰.  The experiments yielding the results with lowest**
**bias and standard deviation are highlighted in blue (EXPTs D and variants).  EXPTs that answer the four specific**
**questions raised in Section 4.4 are indicated.  Tanks used in the calibration are indicated as such and are not independent**
**for that calibration scheme.**

|  | Lab calibration (linear calibration, mole fraction correction) $p_1\ p_0\ c_0\ \chi$ | Daily average of high tank for intercept calibration $p_0$ | Daily average of high and low tanks for linear calibration $p_1\ p_0$ | Daily average high and low tanks mole fraction correction $c_0\ \chi$ | High tank error (‰) mean ± standard deviation for one month (standard error) | Low tank error (‰) mean ± standard deviation for one month (standard error) | Target tank error (independent) (‰) mean ± standard deviation for one month (standard error) |
|---|---|---|---|---|---|---|---|
| EXPT A | - | - | - | - | 0.5±0.4 (0.0) | 1.5±1.3 (0.1) | −0.6±1.3 (0.2) |
| EXPT B | - | - | - | ✓ | Used in cal | Used in cal | −2.2±0.4 (0.1) |
| EXPT C | - | - | ✓ | - | Used in cal | Used in cal | −0.5±1.4 (0.3) |
| EXPT D | ✓ | - | - | ✓ | Used in cal | Used in cal | −0.3±0.4 (0.1) |
| EXPT E (question 1) | ✓ | - | - | - | 0.1±0.4 | 0.0±1.2 | −0.3±1.2 (0.2) |
| EXPT F (question 2) | ✓ | ✓ | - | - | Used in cal | −0.1±0.9 (0.0) | −0.4±0.9 (0.2) |
| EXPT G (question 3) | - | - | ✓ | ✓ | Used in cal | Used in cal | −0.5±1.4 (0.3) |
| EXPT D1 (question 4) | ✓ | - | - | ✓ | Used in cal | Used in cal | −0.3±0.5 (0.1) |
| EXPT D-T (question 4) | ✓ | - | - | ✓ | Used in cal | Used in cal | −0.3±0.6 (0.1) |





**Table 4. Results for the four Marcellus towers using multiple possible calibration schemes. Results for October 2016 are**
**shown for the South, East and North towers. Note that the sample size (used for the calculation of the standard error)**
**was larger for the high and low tanks compared to the target tank, as those tanks were sampled 13.5 times per day.**
**Results for the Central tower are shown for May 2016 (analyzer at manufacturer for repairs during October 2016).**
**EXPT H (only lab calibration) and EXPT D (lab calibration, daily mean of high tank for intercept calibration and daily**
**averages of high and low tanks for mole fraction correction) is as in Table 3. The target tank was independent in these**
**cases. EXPT D-HI-PRE-HI-T is similar to EXPT D, but uses only the high tank value immediately preceding the target**
**tank sampling for the day. The low tank was independent in this case (and only the low tank sampling immediately**
**preceding the target was considered in the results for this case). EXPT D-HI-T used the high tank for the intercept**
**calibration (as in EXPT D), but used the daily average High and the Target tanks for the mole fraction correction. The**
**low tank was again independent in this case. EXPT D-HI-T (highlighted in blue) is the calibration scheme applied to the**
**entire dataset.**

| | Tower | High tank error (‰) mean ± standard deviation for one month (standard error) | Low tank error (‰) mean ± standard deviation for one month (standard error) | Target tank error (‰) mean ± standard deviation for one month (standard error) |
|---|---|---|---|---|
| EXPT H | South | 0.1±0.4 (0.0) | 0.0±1.2 (0.1) | −0.3±1.2 (0.2) |
| EXPT H | East | 0.3±0.4 (0.0) | −0.1±0.8 (0.0) | −0.9±0.9 (0.2) |
| EXPT H | Central | 0.0±0.2 (0.0) | 0.3±0.7 (0.0) | −0.1±0.8 (0.1) |
| EXPT H | North | −0.6±0.4 (0.0) | −1.5±1.3 (0.1) | −1.9±0.6 (0.1) |
| | | | | |
| EXPT D | South | Used in cal | Used in cal | −0.3±0.4 (0.1) |
| EXPT D | East | Used in cal | Used in cal | −0.8±0.5 (0.1) |
| EXPT D | Central | Used in cal | Used in cal | −0.5±0.3 (0.1) |
| EXPT D | North | Used in cal | Used in cal | −0.4±0.7 (0.1) |
| | | | | |
| EXPT D-PRE-HI -T | South | Used in cal | 0.2±0.7 (0.0) | Used in cal |
| EXPT D-PRE-HI -T | East | Used in cal | 0.7±0.6 (0.0) | Used in cal |
| EXPT D-PRE-HI -T | Central | Used in cal | 0.5±0.5 (0.0) | Used in cal |
| EXPT D-PRE-HI-T | North | Used in cal | 0.3±1.3 (0.1) | Used in cal |
| | | | | |
| EXPT D-HI-T | South | Used in cal | 0.2±0.7 (0.0) | Used in cal |
| EXPT D-HI-T | East | Used in cal | 0.7±0.6 (0.0) | Used in cal |
| EXPT D-HI-T | Central | Used in cal | 0.4±0.5 (0.0) | Used in cal |
| EXPT D-HI-T | North | Used in cal | 0.3±1.3 (0.1) | Used in cal |



**Table 5. Possible field calibration tanks and sampling strategies, including those employed in the present study. The**
**"Alternate strategy" column suggests a possible strategy in which two high tanks are employed to achieve the linear**
**calibration for isotopic ratio, and thus laboratory calibration is not required. The compatibility may be improved slightly**
**using this strategy, as both the slope and intercept of the linear isotopic ratio calibration are updated in the field.**
**Furthermore, an independent tank near the ambient range would be preferable for evaluation of performance. In**
**situations when the compatibility requirements are not as stringent (e.g., >0.5 ‰), the strategy detailed in the last column**
**may be appropriate. Sampling times listed exclude transition time between gases.**

| | Present study prior to 3 December 2016 | Present study 3 December 2016 and thereafter | Alternate strategy | Strategy for reduced compatibility requirements (~0.5‰) |
|---|---|---|---|---|
| **Laboratory calibration needed?** | Yes, for linear calibration and mole fraction correction | Yes, for linear calibration and mole fraction correction | No | No |
| **High CH₄ mole fraction tank(s)** | HIGH (10 ppm, −38.3‰, 26 min/day) | HIGH (10 ppm, −38.3‰, 10 min/day) | HIGH (10 ppm, −38.3‰, 4 min/day) | HIGH (10 ppm, −38.3‰, 4 min/day) |
| | - | - | HIGH (10 ppm, −54.5‰, 4 min/day) | - |
| **Low CH₄ mole fraction tanks** | LOW (2 ppm, −23.9‰, 81 min/day) *independent* | LOW (2 ppm, −23.9‰, 54 min/day) *independent* | LOW/SECONDARY TARGET (2 ppm, near −47‰ (ambient), 64 min/day) | LOW (2 ppm, −23.9‰, 6–32 min/day) |
| | TARGET (2 ppm, −47.2‰, 6 min/day) | TARGET (2 ppm, −47.2‰, 54 min/day) | TARGET (2 ppm, near −47‰ (ambient), 64 min/day) *independent* | TARGET (2 ppm, −54.5‰, 6–32 min/day) *independent* |
| **Advantage** | | Reduced noise in calibration due to increased target tank sampling time | Does not require laboratory calibration. Low (secondary target) near ambient range is more accurate reflection of compatibility. | Does not require laboratory calibration. Reduced sampling time of low and target tanks reduces gas usage. Easier to obtain tanks with these specified isotopic ratios. |






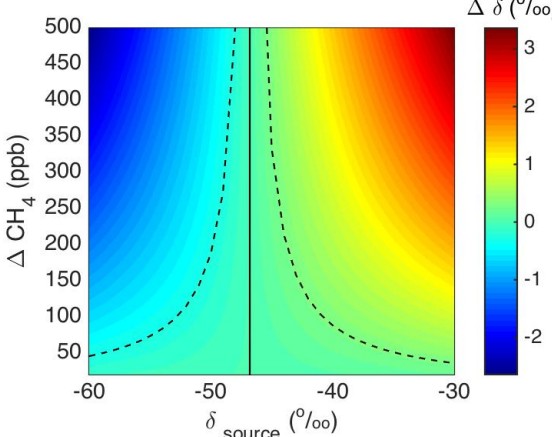

**Figure 1. Isotopic ratio enhancement above background resulting of a mixture of background and source signatures, as a**
**function of source isotopic ratio and $CH_4$ mole fraction enhancement above background. Background CH4 mole fraction**
**was assumed to be 2000 ppb and background isotopic ratio –47 ‰ (vertical solid line). Dashed lines indicate –0.3 ‰ and**
**0.3 ‰ enhancement above background.**





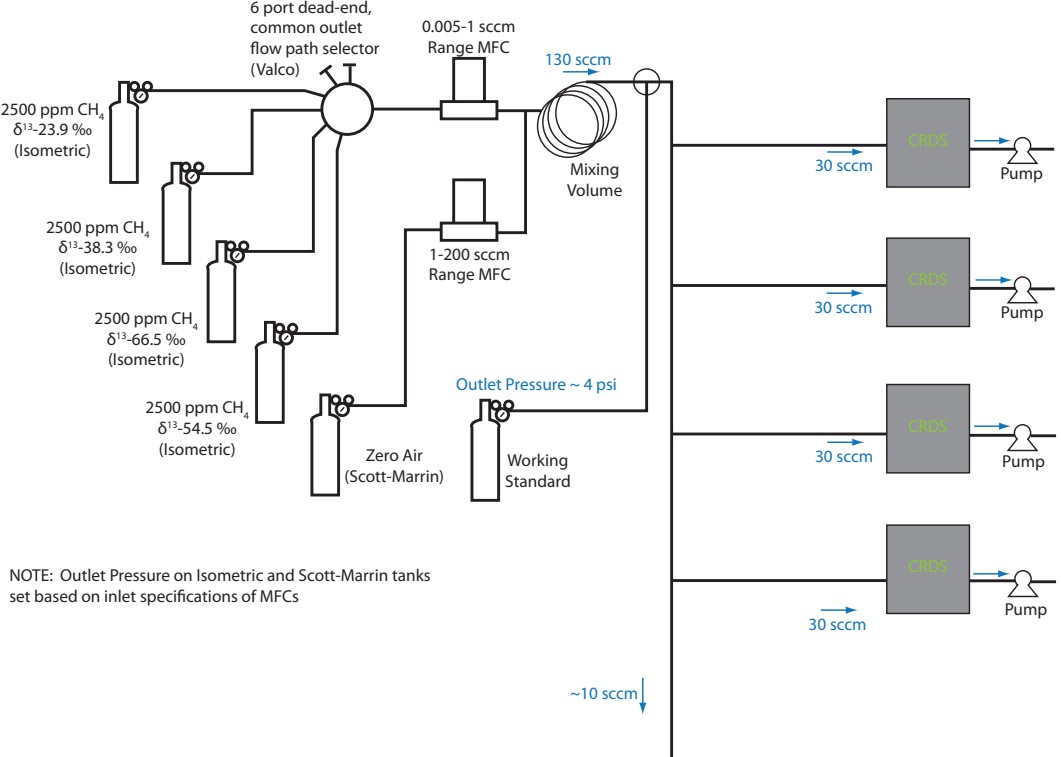

4   **Figure 2. Flow diagram of the experimental setup used for the laboratory calibration of the analyzers.**



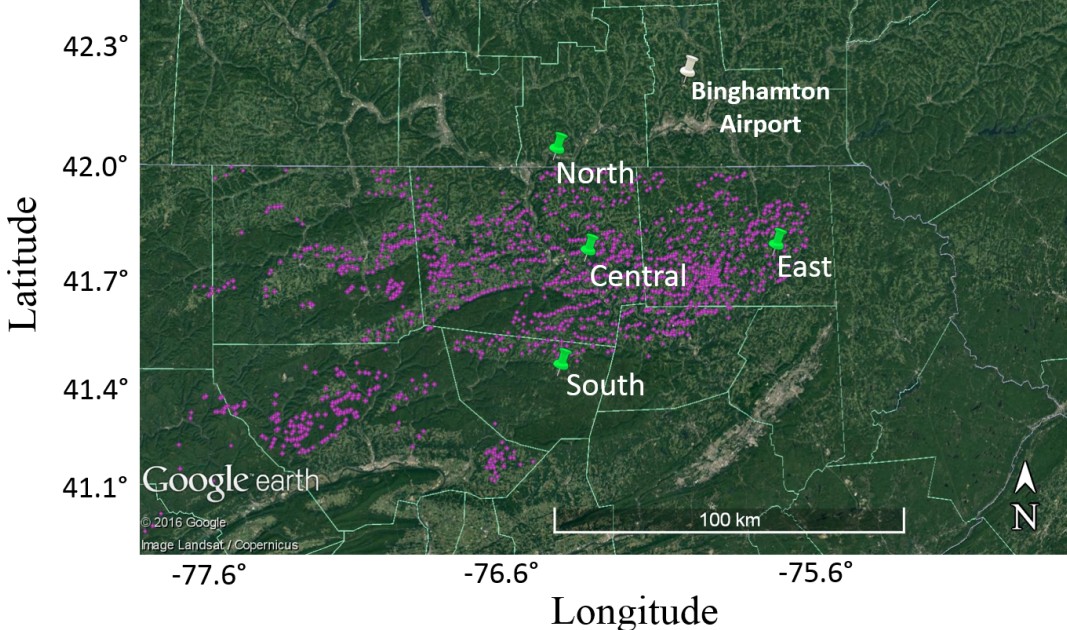

Figure 3. Map of Pennsylvania with permitted unconventional natural gas wells (magenta dots) and network of towers with methane and stable isotope analyzers (Picarro G2132-i). The East and South towers were also equipped with NOAA flask sampling systems. The Binghamton Airport is also indicated.





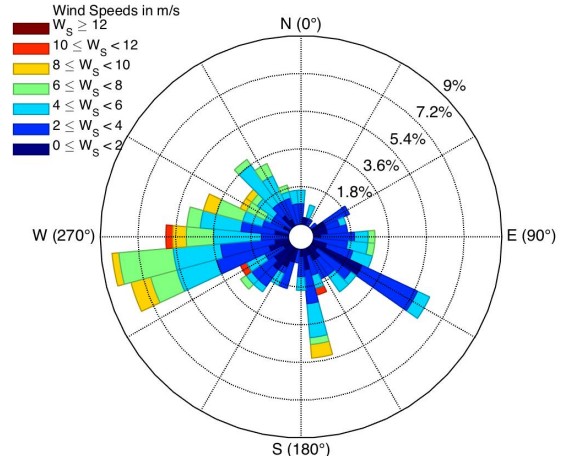

4  Figure 4. Wind rose for surface station at Binghamton, NY airport for the period April 2015 – April 2016 (using the mean
5  of the afternoon hours for each day). The magnitude of wedges indicates relative frequency for each wind direction and
6  the wind speeds are indicated by color.  Calm winds (< 3.6 m s$^{-1}$) are not categorized by direction.


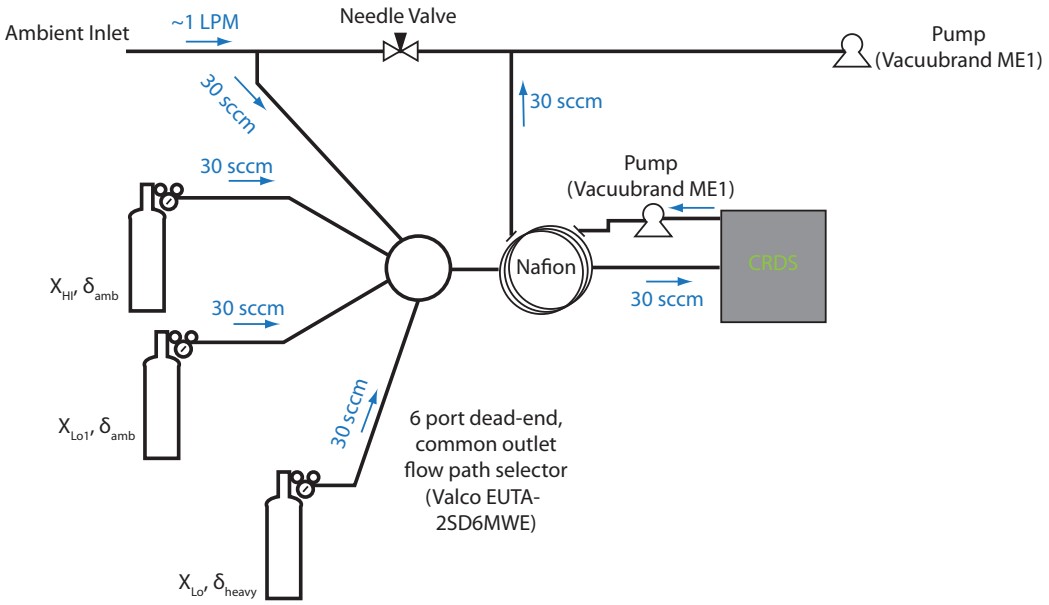

**Figure 5.  Flow diagram of the field calibration system.**





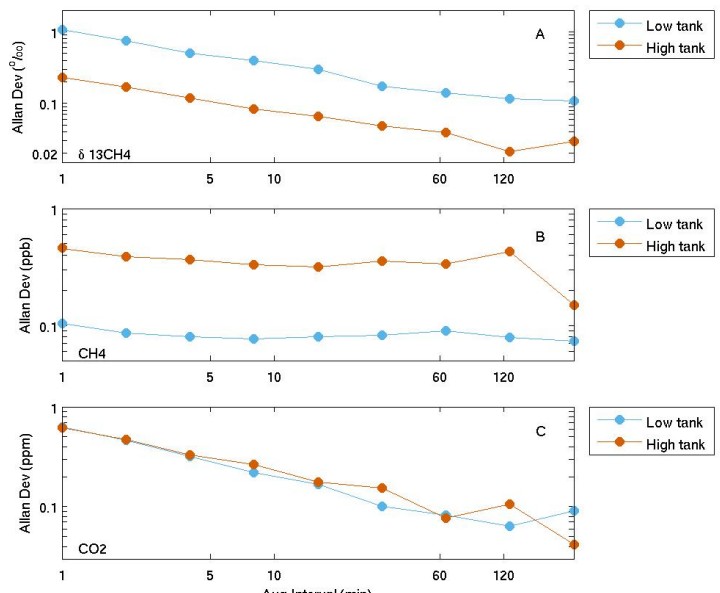

**Figure 6. Allan standard deviation for (A) $\delta^{13}CH_4$, (B) $CH_4$, and (C) $CO_2$ for a high CH4 mole fraction tank (9.7 ppm CH$_4$, ~400 ppm CO$_2$, –38.3 ‰ $\delta^{13}CH_4$) (orange) and a low (1.9 ppm CH$_4$, ~400 ppm CO$_2$, –23.7 ‰ $\delta^{13}CH_4$) tank (blue). The x-axis is truncated to focus on minimum averaging times required to achieve the desired compatibility goals.**





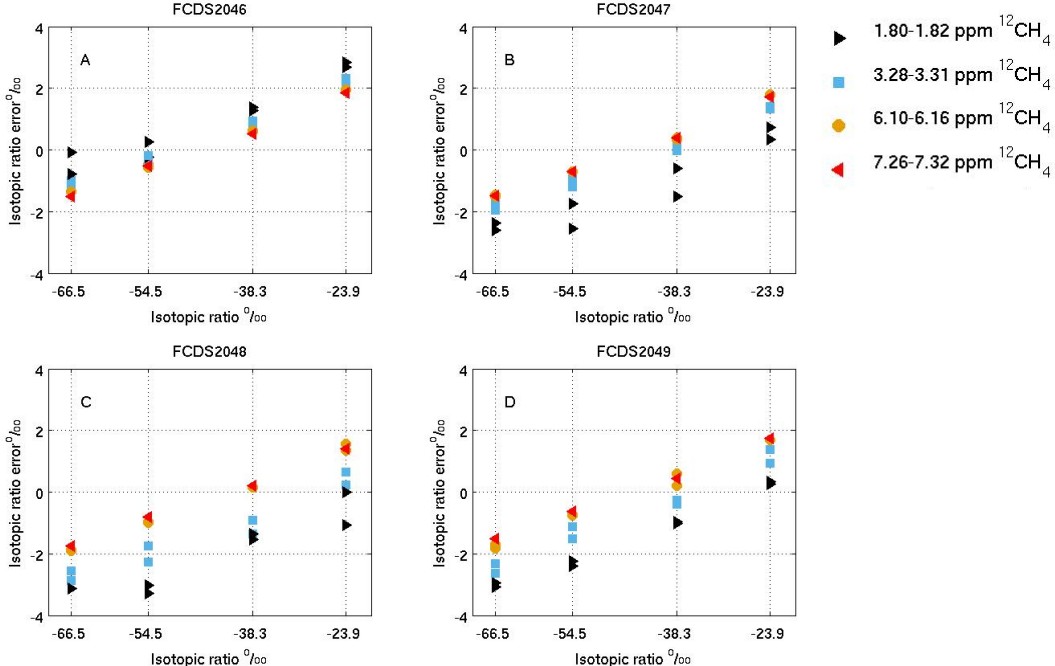

**Figure 7. Isotopic ratio error as a function of isotopic ratio for each of the four analyzers (A – D). The colors indicate the $^{12}CH_4$ mole fraction, as shown in the legend. The serial numbers (FCDS2046, FCDS2047, FCDS2048, and FCDS2049) of the analyzers are indicated as well. These analyzers were deployed at the South, Central, North and East towers, respectively. Interpolating from the Allan standard deviation results (Fig. 6), the estimated precision is 0.40 ‰ for the 1.80–1.82 ppm CH$_4$ tests, 0.34 ‰ for 3.28–3.32 ppm CH$_4$ tests, 0.24 ‰ for 6.10–6.16 ppm CH$_4$ tests, and 0.20‰ for 7.26–7.32 ppm CH$_4$ tests. The precision could be improved by averaging over longer periods.**



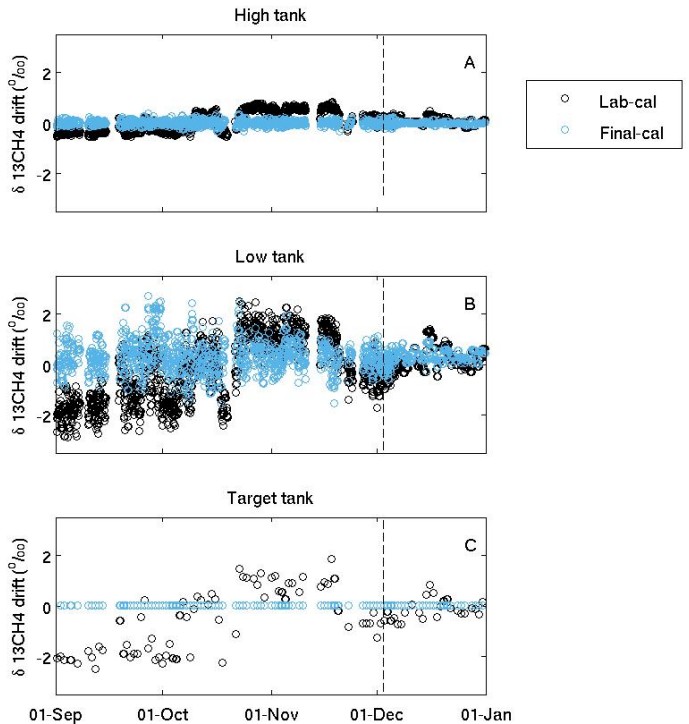

**Figure 8.  Results following isotopic ratio laboratory calibration only (black) and following the EXPT-D-HI-T protocol**
**(blue) for the South tower for September - December 2016 for the "high" CH$_4$ mole fraction tank (A), "low" CH$_4$ mole**
**fraction tank (B), and target tank (C).  The low tank was independent of the isotopic ratio calibration.  An improved**
**calibration tank sampling strategy was implemented on 3 December 2016 (indicated by vertical dashed lines).**



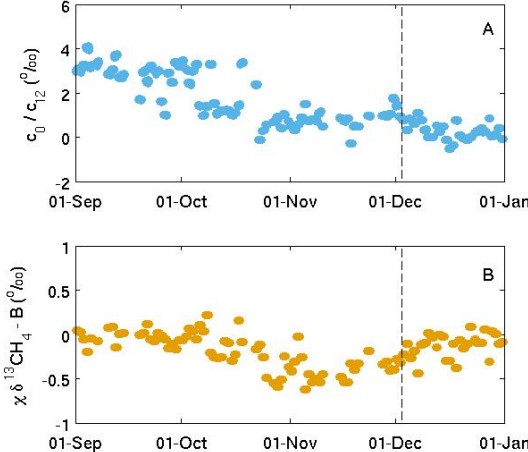

**Figure 9.** Effect of each of the calibration coefficient terms for the South tower for September - December 2016 for the optimized calibration scheme. The terms $c_0$ (A) and $\chi$ (B) in Eq. (3) are time-dependent drift terms. These parameters vary because of spectral variations in the optical loss of the empty cavity ($c_0$), and because of errors in the temperature or pressure of the gas, or changes in the wavelength calibration ($\chi$). Note the differing scales. An improved calibration tank sampling strategy was implemented on 3 December 2016 (indicated by vertical dashed lines).



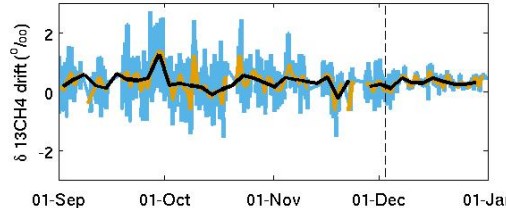

**Figure 10.  Low tank methane isotopic ratio differences from known value, for the individual calibration cycles (blue),**
**and for 1-day (red) and 3-day (black) means of the calibration cycles, for the South tower for September – December**
**2016.  An improved calibration tank sampling strategy was implemented on 3 December 2016 (indicated by the vertical**
**dashed line).  The low tank is independent of the isotopic ratio calibration.**





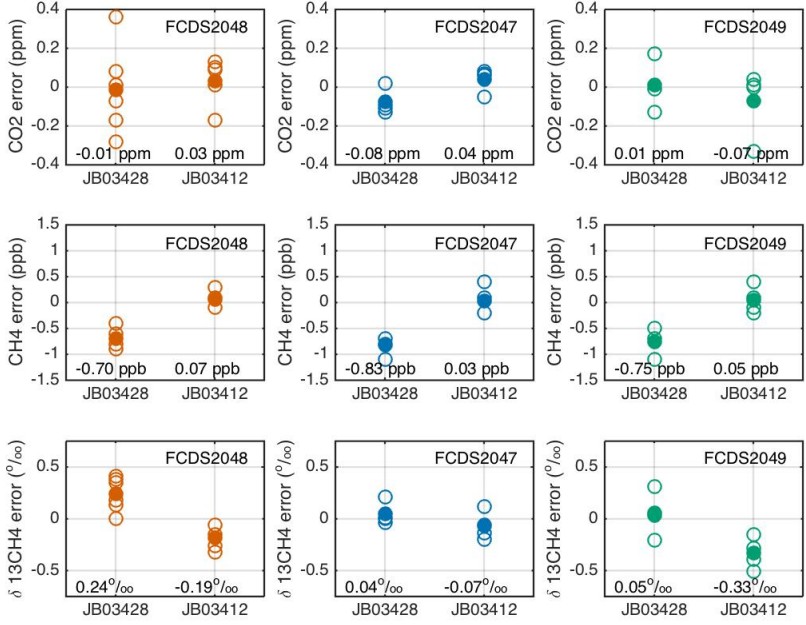

**Figure 11. Results from round-robin style testing using two NOAA/INSTAAR tanks (JB03428: –46.82 ‰ $\delta^{13}CH_4$, 1895.3**
**ppb $CH_4$ and 381.63 ppm $CO_2$; and JB03412: –45.29 ‰ $\delta^{13}CH_4$, 2385.2 ppb $CH_4$ and 432.71 ppm $CO_2$) for $CO_2$ (top row),**
**$CH_4$ (middle row), and $\delta^{13}CH_4$ (bottom row), for the analyzer deployed at the North tower (serial number FCDS2048; left**
**column), at the Central tower (serial number FCDS2047; middle column), and at the East Tower (serial number**
**FCDS2049; right column). These tests were completed in the laboratory, post deployment (March 2017). The analyzer**
**deployed at the South tower (serial number FCDS2046) was not included in these tests. Open circles are individual tests**
**and filled circles are the means of the individual tests for each analyzer/constituent. The mean error for each**
**analyzer/tank/constituent is indicated in the plots.**




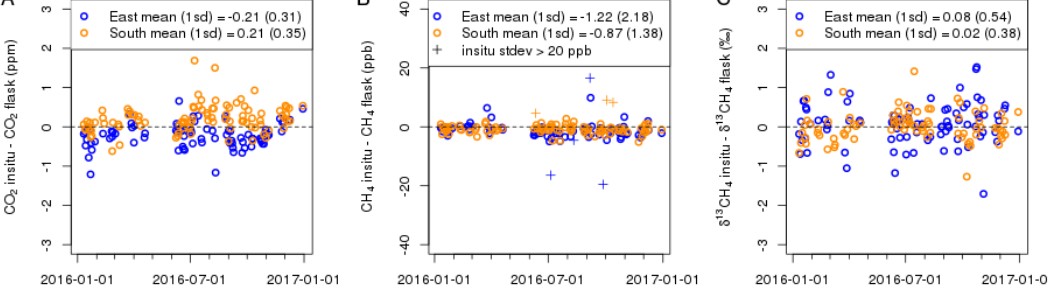

**Figure 12.  Afternoon in-situ to flask differences for January – December 2016 for the East (blue) and the South towers**
**(orange) for A) $CO_2$, B) $CH_4$, and C) $\delta^{13}CH_4$.  For $CH_4$, data points with high temporal variability (standard deviation of**
**raw ~2sec data within the hour > 20 ppb) are indicated by '+' symbols and have been excluded.  The standard deviation**
**of the in-situ to flask differences are shown in parentheses on each plot.  The standard errors, indicating an estimate of**
**how far the sample mean is likely to be from the true mean, is 0.24 ppb $CH_4$, 0.03 ppm $CO_2$ and 0.06 ‰ at the East tower**
**and 0.14 ppb $CH_4$, 0.04 ppm $CO_2$ and 0.04 ‰ at the South tower.**





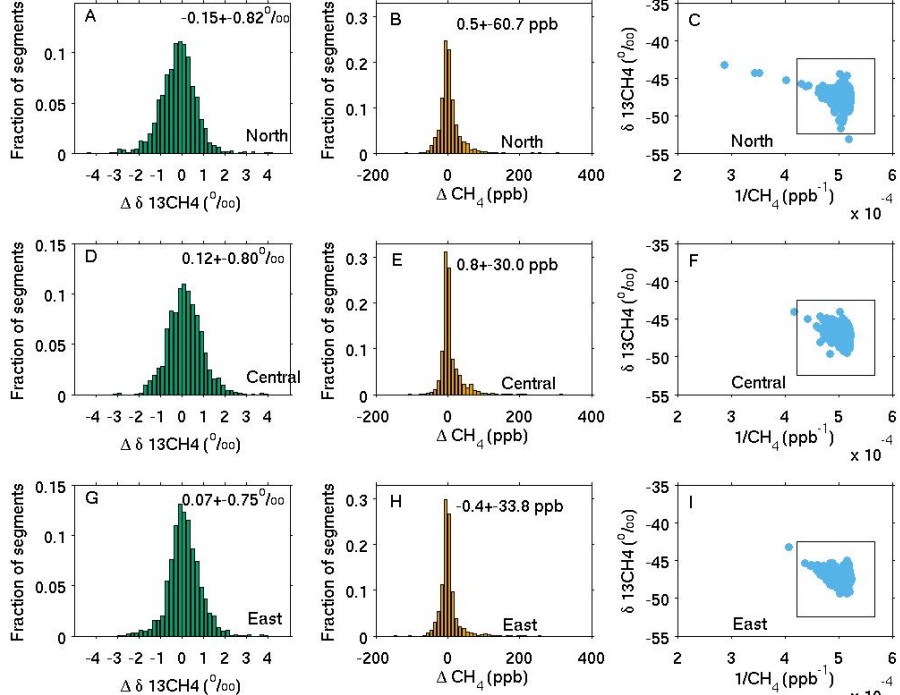

**Figure 13. Probability distribution function of isotopic ratio enhancement above the background South tower for the A)**
**North, D) Central, and G) East towers for afternoon hours (1700–2059 UTC, 1200–1559 LST). The time scale of the**
**individual data points for all plots is 10 min and the time period is January – May 2016. The bin size for A), D) and G) is**
**0.2 ‰. Probability distribution function of methane mole fraction enhancements for the B) North, E) Central, and H)**
**East towers. Note that the scale for B), E, and H) has been truncated to focus on majority of the data points. The bin size**
**is 10 ppb $CH_4$. Keeling plots for the C) North, F) Central, and I) East towers. The black box in each plot indicates the**
**approximate scale of the corresponding isotopic ratio enhancement and methane mole fraction enhancement plots.**



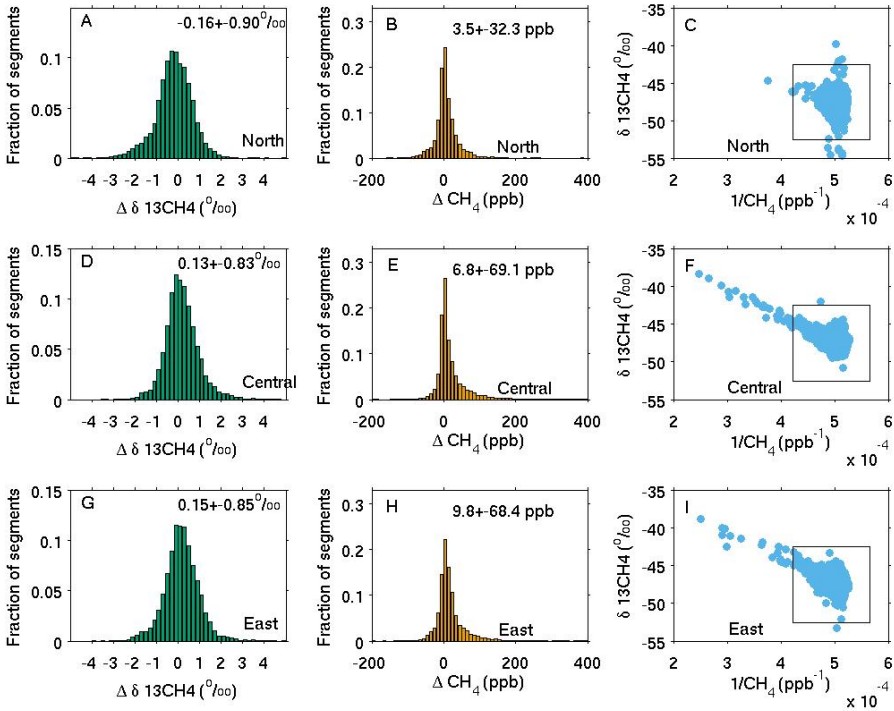

**Figure 14.  Probability distribution function of isotopic ratio enhancement above the background South tower for the A)**
**North, D) Central, and G) East towers for all times of data excluding the afternoon hours shown in Fig. 13.  The time**
**scale of the individual data points for all plots is 10 min and the time period is January – May 2016.  The bin size for A),**
**D) and G) is 0.2 ‰.  Probability distribution function of methane mole fraction enhancements for the B) North, E)**
**Central, and H) East towers.  Note that the scale for B), E), and H) has been truncated to focus on majority of the data**
**points.  The bin size is 10 ppb CH₄.  Keeling plots for the C) North, F) Central, and I) East towers. The black box in each**
**plot indicates the approximate scale of the corresponding isotopic ratio enhancement and methane mole fraction**
**enhancement plots.**





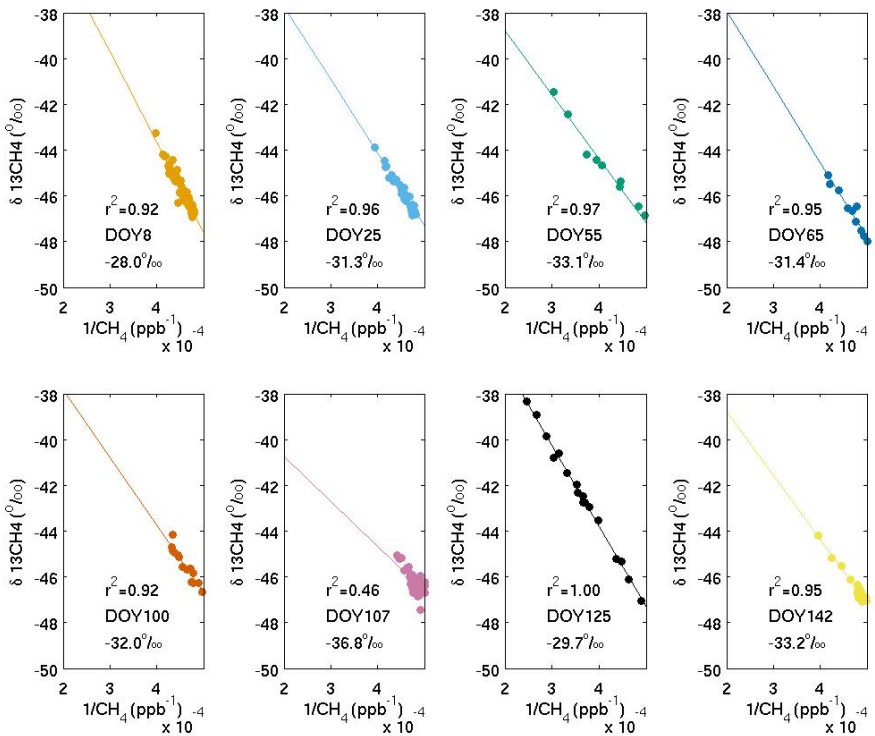

2  **Figure 15. Keeling plots for the Central tower for the eight largest peak in the non-afternoon methane time series. Black**
3  **lines indicate the best-fit lines. Correlation coefficients ($r^2$), day of year (DOY) and y-intercepts are indicated in the plots.**



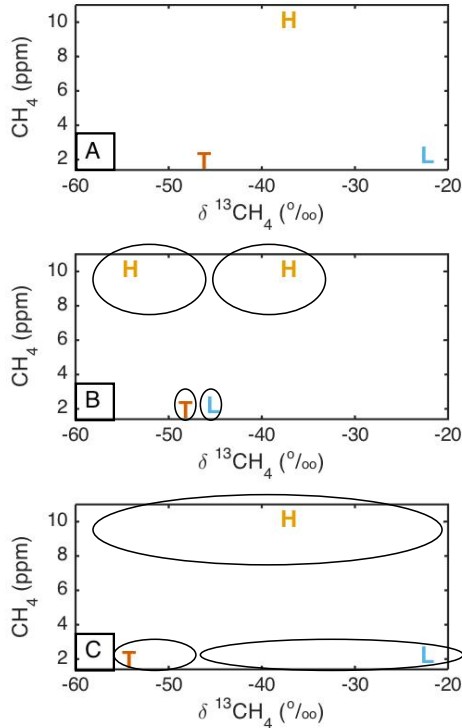

Figure 16. Graphical representation of the field calibration tanks used in the present study (A), for an alternate strategy
(as in Table 5) (B), and for a strategy for reduced compatibility requirements (C). Orange 'H' symbols indicate high mole
fraction tanks, blue 'L' symbols indicate low mole fraction tanks, and red 'T' symbols indicate target tanks, as in Table 5.
In A), the low tank is independent, but in B) and C), the target tank is independent. A "Target" tank is commonly used to
indicate an independent tank. The ovals around the tanks in B) and C) indicate desirable ranges of values. Ideally,
calibration tanks are near the ambient values to be measured, but in this case, specific values are more easily obtained (–
54.5 ‰, –38.3 ‰, –23.9 ‰, from Isometric Instruments, Inc.)

