# Peer review of "Calibration and Field Testing of Cavity Ring-Down Laser"

_Atmospheric Measurement Techniques, 2017_

## Referee Comment (RC1) · D. Lowry (Referee) · 10 Nov 2017

General Comments

This manuscript discusses a specific question within the scope of AMT, presenting new data on the difficult task of calibrating laser methane isotope instruments at unmanned tower sites.

The methods are outlined in extensive detail, but this also highlights how a little more consideration of the isotopic signatures of influencing methane sources at the start of

the project could have saved significant time spent later refining calibration routines. The choice of enriched calibration mixtures over deleted ones suggests an expectation of 13C-enriched sources.

In places the manuscript is quite difficult to follow as there are frequent references to previous or subsequent sections. There are also a lot of small errors that need tidying up, and some points that need greater clarification. This includes the selection procedure of points for the individual peak Keeling plots. Some of the subsequently 'unused' EXPTs could be removed from the text (but remain in the table), as they are not used in the revised calibration routine.

The manuscript also lack a good final summary, possibly due to the conference volume time constraints, but it would be good to see the following questions discussed at the end: 1) In hindsight what could have been done differently? 2) What are the recommendations for anyone else undertaking the set-up of a similar network? 3) What are the limitations and advantages of the CRDS technique at fixed tower sites compared with IRMS at a similar type of site (e.g. Rockmann et al., 2016)?

The manuscript also highlights the need for suitable isotopic standards for urban / source region standards within this community, as the measured isotopic ratios fall between those commonly measured at source, or the tight constraints around -47 ‰ needed for background sites. Something like -52 and -42 ‰ at 3 ppm mole fraction are toward the limits of the enhancements measured at such sites, and so would be very useful as standards.

Detailed Comments

Abstract (see also later comments)

Page 1 Line 24 – Why only calibration with high methane mole fraction air bottles?

Page 2 Line 2-5 - This technique might work here because the Marcellus gases are significantly enriched in 13C, but in many regions (eg. Australia, much of the EC) there

are no major sources enriched in 13C relative to atmospheric background, so that the sources would be very difficult to distinguish far away from the point of emission.

Page 2 Line 4 – What is the error on the -31.2 ‰ The literature suggests a range of source values.

Main Text

Page 3 Line 18 – t missing from Schweizke

Page 5 Lines 16-22 – yes it should be symmetrical at 35 and -60 ‰ around a background of -47.5 ‰ but the colours on Fig 1 don't show an even change to enriched and depleted values around the background composition (see also comment on Figure).

Page 6 Line 3 – the noise is known to be less for higher mole fractions – please cite the source of this

Page 6 Line 24 – slight clarification needed – was 1 sccm for the standard line and 500 sccm for the zero air line

Page 7 Lines 1-7 – it would be useful here to include the flow rate through the instrument to give some idea of how much cal gas was being used.

Page 7 Lines 19-32 – some of the terms are not clearly explained here (for example Bdefault) so please either clarify or remove and refer the reader directly to Rella et al. (2015). As it would be good to be able to refer to the calibration plot at this point to see the isotopic offset compared to known values for each standard at different mole fractions can Fig.7 go here, or be linked to section 4.2 or at least referred to?

Page 8 Lines 3-10 – The study area section and accompanying figures seem a little out of place here as the sites are not discussed for a long time. Is there a better position for this?

Page 8 Line 16 – not been demonstrated for laser instruments, for IRMS see Rockmann et al.(2016).

Page 8 Line 25 – using the word 'sampling' for the measurement of standard cylinders is easy to confuse with the actual measurement of ambient air at the towers. Is there an alternative word that could be used for this procedure?

Page 8 Lines 26-27 – the reasoning behind choosing the mole fractions for tanks with these isotopic standards is not clear here, and is even less clear later when the -24 ‰ cylinder is dropped from the calibration routine.

Page 9 Lines 11-13 – what is a large flow rate and what is the delay time for air entering the inlet to instrument measurement?

Page 9 Lines 28-30 – should the aim not be to achieve the best possible precision, not just to reach target compatibility? Would the high mole fraction tanks not achieve better precision if analysed for 32 minutes? If the shorter times are chosen just to save cal gas then it should be made clear somewhere.

Page 10 Lines 18-24 – the -24 ‰ standard at any concentration was never going to be useful at the tower sites as it is even beyond the range of sampling a pure source emission. A clear understanding of the maximum mole fractions and isotopic shifts observed at the towers would have been useful before production of the working standards for these sites, even if it meant not using the first 6 months of isotope data.

Page 13 Lines 26-27 – what are the uncertainties on the NOAA tanks?

Page 14 Lines 8-10 – replace tubing with tube, and on-board with internal

Page 14 Section 5.3 – what is the relevance of this section? Unless the results of this test are shown somewhere, and referred to here, then is this section necessary for the points under discussion?

Page 14 Line 16 – 14:00-15:00 isn't late afternoon

Page 14 Line 18 – do you mean measured? Either 'samples were collected' or 'flasks were filled'

Page 15 Lines 9-10 – 'An error in isotopic ratio as a function of isotopic ratio' is not very informative. Figure 7 shows that there is an offset in the measured isotopic ratio as a function of the changing known isotopic ratio, which seems to be quite constant for all instruments at higher mole fractions, but instrument-specific at near-background mole fractions.

Section 6.2 – I initially thought that the experiments involved changing over to different cylinders, which seems to be the case when changing to the new calibration routine, but most of these EXPTs seem to be just manipulation of the data for different standards, so please can you make clear at the start of this section that these are mostly changes in the calibration calculations and not changes to the cylinders being analysed. Some of the results of these experiments are not used as they do not improve the required precisions. Do these need to be described in the text?

Page 16 Line 13 – a result is a result and cannot be changed. Do you mean 'improve the calculated precision'?

Page 18 Line 16 – 0.18 ‰ is stated above this as the daily average, not the hourly average as used here.

Section 6.6 – the side-by-side testing results are mentioned here, but have a lot of content overlap with the methodology for side-by-side testing. Does this have to be in two places? It isn't particularly relevant to the core isotopic story.

Page 20 Line 2 – 'mean flask' – is this referring to the rapidly-filled or the hour-long filled flasks?

Page 20 Line 5 – 'hourly flask to in-situ differences for the year'. It isn't clear what this means

Page 20 Line 19 – 'The time scale of the individual data points was 10 min'. Do you mean averaging interval?

Page 20 Lines 32-33 – It isn't clear why 'enhancements greater than 6ppb CH4 in

magnitude' are '3 times the target compatibility of 0.2 ‰ ÌĄ.

Page 21 Line 15 – mean of -31.2± ? ‰

Page 21 Lines 21-23 – this section needs sorting out; firstly they are not peak heights but enhancements over background, and it is a 2.5 to 8.7 ‰ positive shift in measured isotopic ratio. What does ' reduced methane enhancement at other data points within the peak' mean? If the maximum enhancement is 2008 ppb, why does it mention 1500 ppb maximum earlier?

Page 22 Line 21 – remove the s from fractions

Page 22 Line 26 – replace improving with improved

Page 22 Lines 35 to 37 – given the availability of the Isometric flasks, my preference for a calibration would be to create a high and low mole fraction cylinder from both of the -54 and -38 ‰ standards, used in combination with a low ambient cylinder and a similar target gas, or at least a high and low at -38 ‰ as that is the direction your sources are taking the ambient mix. More cylinders, but it should improve the correction in the triangle of 13C-mole fraction space where the measurements lie.

Page 23 Line 30 – an isotopic ratio enhancement of -0.6 ‰Ṫwo problems with this; surely this should be a + and not a -, and how do you enhance a ratio? Normally this would be heavy or light for a change in ratio as 13C increases or decreases, or enrichment or depletion if talking about the individual 13C, so it does represent an enrichment in 13C.

Page 24 Line 15 – already presented so should be ' we have presented.

Page 24 Lines 15-19 – this is a rather abrupt ending – see general comments for suggestions of what to add in summation.

References

All those present seem to be correct in the text, just move the date on P27 Line 7 up a

line.

The following references are mentioned in the text but are not in the list: Conway et al., 2011 Montzka et al., 2011 Turnbull et al., 2012 Vaughn et al., 2004

Tables

Table 2 – Not clear what this table is representing from the caption? Are these due to interferences with the 12CH4 and 13CH4 spectral lines? What is the maximum CH4 at these sites and does this increase or decrease the interference on each of these species?

Table 3 caption Line 10 – should be at not and

Table 5 – the alternative strategy looks to be the best, but given the observed range of measurements up to 4 ppm, would it not be better to correct with a lower precision at 4 ppm than a calculation of the offset from 10 ppm? You have all of the Isometric standards available, so 'easier to obtain tanks with these ratios' is not an advantage of the reduced compatibility option, but all of the options.

Figures

Is there a way to sort out the superscript and subscripts on the vertical axes of the graphs or is this a software limitation?

Figure 1 - It needs to be mentioned in the caption what are the signatures of the isotopic end members used to create this plot. There is an uneven spread of colour change around the -47 ‰ point, which suggests that this has been calculated with -30 and -60 ‰ source increments and not the -35 and -60 ‰ mentioned in the text.

Figure 2 caption – mention the cylinder volumes

Figure 4 – The lowest category of wind direction is 0-2 m/s, but the caption states that calm winds below 3.6 m/s are not categories by direction, so two things to correct here. 3.6 m/s is not calm. I think that this is a scale conversion problem from km/h, and calm

[Figure]

should be either less than 1 m/s (in which case the lower category displayed should be 1-2 m/s), or <0.1 m/s. It isn't clear why Figs 3 and 4 are so early in the manuscript. Until the tower location data is discussed there isn't really the interest in seeing these influences.

Figure 5 – What is the flow rate of the pump on the exhaust line (ME3)? It looks like it should be balancing out the ambient inlet flow.

Figure 6 caption – incorrect 13C shown on first line

Figure 7 – The residual errors seem to be quite consistent between instruments for the higher mole fraction cylinders at all sites. The correction procedure for the small number of elevated mole fraction samples is not very obvious from the text.

Figures 8 and 9 – what are the error bars for these data points? If they cannot be added to the graphs can they be alluded to in the caption?

Figure 11 – Can North, Central and East be labeled at the top of each column?

Figure 13 – add measured before isotopic in the first line of the caption. The big delta small delta use on the left column horizontal axes is not explained. The x104 used in the right column needs correctly positioning, although it would be much clearer just using 0.2 to 0.6 in 1/CH4 ppm.

Figure 14 – the 15 ‰ spread of isotopic ratios measured at near background mole fractions reduces confidence in the data. The text suggests that there is a gradual improvement in measurement precision as the mole fraction increases. The Keeling plots suggest that the transition is quite sharp, and the isotope CRDS that I have seen in field operation seem to have a sharp improvement at 7-8 ppm CH4, so can this be clarified in the caption or elsewhere in the text.

Figure 15 – the Keeling plots are good but indicate very little variation in measured isotopic ratio for a given mole fraction even at near background mole fractions. Are all of the points from these peaks used? It would be good to see the actual time graph

[Figure]

of the isotopes for 1 of these graphs to see the points that have been selected for use in the Keeling plot. The source intercept calculation is without errors. Can these be calculated

---

## Referee Comment (RC2) · Anonymous Referee #2 · 7 Dec 2017

Review of the manuscript: "Calibration and Field Testing of Cavity Ring-Down Laser Spectrometers Measuring CH4, CO2, and $\delta$13CH4 Deployed on Towers in the Marcellus Shale Region", submitted to Atmos. Meas. Tech. , by Natasha Miles et al.

The paper is describing the measurements of atmospheric mole fractions of $\delta$13CH4, CH4 and CO2 at four sites in Pennsylvania. More precisely the manuscript describes the optimization of the technical setup based on lab and field tests.

The manuscript needs to be reorganized to reduce the back and forth between test

descriptions and their results, which makes the reading quite difficult. There are too many redundancies, and unclear statement. When doing that I also suggest to shorten the manuscript. Some conclusions appear obvious, like for example the statement that field calibrations significantly improved the measurements compatibility. Also the so-called optimal calibration strategy refers to the design which was decided a priori and slightly modified during the campaign, but there was no plan to really evaluate alternative design. The conclusion should be written in a more concise way, focusing on the recommendations gained from the experiment.

Introduction: the introduction need to be reorder in order. For example the first paragraph of page 4 describing the interest of tower versus aircraft, appears between two paragraphs discussing more technical points about CRDS measurements Page 4 / Line 13: "three field calibration tanks. . .": I would rather say two calibration tanks plus one target tank used as quality control and not used in the calibration. Allan variances tests; calibrations tests (Page 6 / Line 31): there are many back and forth between description of the set up and the results, which confuse the paper. Page 9: In-situ field calibration: is the Nafion required for the setup ? Have you compared possible biases due to the use of the Nafion versus the water vapor correction ? I am not fully convinced by the strategy of humidifying the dry calibration tanks. Page 9: 4 min flushing: how do you estimate those 4 minutes as sufficient for the flushing ? Page 12: background site: why don't you select the background site as a function of wind direction rather than picking up one site for the full period ? Page 14: Allan results: For CH4 and CO2 it should be noted that the results seem to be not as good as the performances obtained with G1301/G2401 analyzers. Do you know the reason which could explain a difference of the performances between those analyzers ? Page 16: Calibration scheme: the presentation of the different tests should probably be shortened. Is there a difference between Expt E and H designs ? I would appreciate an evaluation of the optimum frequency of the field calibration sequences (intermediate between 0 and once per day). From the variabilities shown on Fig.8 and 9 it looks like a reduction of the calibration frequency to once every few days would not affect

by much the measurements. Fig.8: the legend is misleading since the so-called target tank is used as a calibration tank. To make it clear you should add comments in the legend of each figure (e.g. Target tank (used as CAL)) Page 17: Fig. 9B and 9C should rather be 9A and 9B Page 20 Line 27: suppress 'For the daily afternoon averages,'. Not clear what you mean by a 'reduction' of 0.6-0.7pmil. Page 20 Line 32: Why do you compare CH4 enhancements (6ppb) with 13CH4 target compatibility (0.2pmil) ? Page 21: lines 22/23: Unclear statement about the dilution of local source. The discussion about the source signature need to be clarified, or preferably merged in the discussion section. Page 23: lines21/22: unclear statement. Conclusion: in your last sentence I would like to see also a comment or discussion that the strategy of using continuous measurements at four tower is maybe not the optimal one for the quantification of such sources.

Please also note the supplement to this comment:
https://www.atmos-meas-tech-discuss.net/amt-2017-364/amt-2017-364-RC2-supplement.pdf

---

## Author Response (AR1)

Dear Editors,

In light of the reviewer comments, we have reorganized the manuscript to be organized by topic rather than having separate methods and discussion sections. Thus back and forth references have been dramatically reduced. We also simplified the manuscript by eliminating discussion of calibration methods that were not actually used. We made clear recommendations for similar future networks, e.g., the isotopic ratio of the field tanks, and the lab and field tank sampling times. Our responses to the reviews are in-line, in blue below.

Thank you,

Natasha Miles

General Comments

This manuscript discusses a specific question within the scope of AMT, presenting new data on the difficult task of calibrating laser methane isotope instruments at unmanned tower sites.

The methods are outlined in extensive detail, but this also highlights how a little more consideration of the isotopic signatures of influencing methane sources at the start of the project could have saved significant time spent later refining calibration routines. The choice of enriched calibration mixtures over deleted ones suggests an expectation of 13C-enriched sources.

Thank you for your thoughtful comments. We shortened the paper by removing most of the discussion of choosing the optimal calibration study. Instead we described how we actually processed the data, rather than describing all the options we explored. We agree that the choices of field tanks and the field tank testing times were not ideal and have added specific recommendations for future similar networks to the paper.

In places the manuscript is quite difficult to follow as there are frequent references to previous or subsequent sections. There are also a lot of small errors that need tidying up, and some points that need greater clarification. This includes the selection procedure of points for the individual peak Keeling plots. Some of the subsequently 'unused' EXPTs could be removed from the text (but remain in the table), as they are not used in the revised calibration routine.

We have addressed these concerns by reorganizing the bulk of the paper. Instead of strictly separating methods and results, we have now inter-mixed these, with the paper being organized by topic. We believe this does enhance readability of the paper. While not the traditional method of manuscript organization, it is sometimes used (e.g., Rella et al. 2015). We also added an example of a time series used for the individual peak Keeling plots, as suggested. We removed lengthy discussion of calibration procedures not actually used to process the data.

The manuscript also lack a good final summary, possibly due to the conference volume time constraints, but it would be good to see the following questions discussed at the end: 1) In hindsight what could have been done differently? 2) What are the recommendations for anyone else undertaking the set-up of a similar network? 3) What are the limitations and advantages of the CRDS technique at fixed tower sites compared with IRMS at a similar type of site (e.g. Rockmann et al., 2016)?

We have reworked the final section to include recommendations for future similar deployments and have mentioned the potential of high-temporal-resolution methane isotopic ratio data, when combined with modeling, to constrain regional methane budgets. Previously the utility of networks of such data was not mentioned.

The manuscript also highlights the need for suitable isotopic standards for urban / source region standards within this community, as the measured isotopic ratios fall between those commonly measured at source, or the tight constraints around -47 ‰ needed for background sites. Something like -52 and -42 ‰ at 3 ppm mole fraction are toward the limits of the enhancements measured at such sites, and so would be very useful as standards.

We included recommendations for future work with these thoughts in mind.

Detailed Comments

Abstract (see also later comments)

Page 1 Line 24 – Why only calibration with high methane mole fraction air bottles?

We edited that sentence to read, "Prior to deployment, each analyzer was tested using bottles with various isotopic ratios, from biogenic to thermogenic source values, which were diluted to varying degrees in zero air, and an initial calibration was performed." We were referring to the Isometric Instruments bottles (as high methane mole fraction air bottles). We did only use the high mole fraction mixture in the initial calibration prior to deployment, because the field calibrations supersede the correction of the mole fraction dependence of the isotopic ratio (and because we didn't measure the low mole fraction mixtures long enough). Those details are in the text, but are too detailed to describe in the abstract. Thank you for pointing out the confusing statement in the abstract.

Page 2 Line 2-5 - This technique might work here because the Marcellus gases are significantly enriched in 13C, but in many regions (eg. Australia, much of the EC) there are no major sources enriched in 13C relative to atmospheric background, so that the sources would be very difficult to distinguish far away from the point of emission.

Good point. We added a sentence to the conclusions, "We note that the Keeling plot approach to determine source isotopic signatures far from the point of emission will be difficult to apply in regions without sources that are significantly depleted or enriched in 13CH4 compared to ambient."

Page 2 Line 4 – What is the error on the -31.2 ‰ The literature suggests a range of source values.

We added the standard deviation of values that we determined and edited the last phrase to read, " within the wide range of values consistent with a deep-layer Marcellus natural gas source. "

Main Text

Page 3 Line 18 – t missing from Schweizke

Corrected.

Page 5 Lines 16-22 – yes it should be symmetrical at 35 and -60 ‰ around a back- ground of -47.5 ‰ but the colours on Fig 1 don't show an even change to enriched and depleted values around the background composition (see also comment on Figure).

Yes, thank you for pointing out this error. We have corrected the calculations for the figure to reflect -35 and -60‰, with a background of -47.5‰ used. We added further description of the end members to the caption, as requested.

Page 6 Line 3 – the noise is known to be less for higher mole fractions – please cite the source of this

We added the reference for this (Rella et al., 2015).

Page 6 Line 24 – slight clarification needed – was 1 sccm for the standard line and 500 sccm for the zero air line

Yes, and we have added that clarification.

Page 7 Lines 1-7 – it would be useful here to include the flow rate through the instru- ment to give some idea of how much cal gas was being used.

We noted in this section: "With the flow rate of 0.400 sccm for the isotopic standard bottles, the total volume of standard gas used was 88 cc." Here we focused on the standard bottles rather than the zero air, since the zero air is inexpensive. We also added in Section 4.3 (In-situ field calibration gas system), "The flow rate of the instruments was 35 cc/min, and the 150A tank size was used, corresponding to $4.021 \times 10^6$ cc at standard pressure and temperature. Thus there was gas sufficient for 9800 calibration cycles, or 725 days at this calibration frequency."

Page 7 Lines 19-32 – some of the terms are not clearly explained here (for example Bdefault) so please either clarify or remove and refer the reader directly to Rella et al. (2015).

We added a description of the meaning of the terms B and Befault.

As it would be good to be able to refer to the calibration plot at this point to see the isotopic offset compared to known values for each standard at different mole fractions can Fig.7 go here, or be linked to section 4.2 or at least referred to?

With our reorganization of the paper, the figure is included in the mentioned section.

Page 8 Lines 3-10 – The study area section and accompanying figures seem a little out of place here as the sites are not discussed for a long time. Is there a better position for this?

We moved the section describing the study area (including the figures originally numbered 3 and 4) to just prior to the section describing "Methods for determining enhancements".

Page 8 Line 16 – not been demonstrated for laser instruments, for IRMS see Rockmann et al.(2016).

We added "CRDS" to the sentence.

Page 8 Line 25 – using the word 'sampling' for the measurement of standard cylinders is easy to confuse with the actual measurement of ambient air at the towers. Is there an alternative word that could be used for this procedure?

We changed this to 'testing' throughout the text.

Page 8 Lines 26-27 – the reasoning behind choosing the mole fractions for tanks with these isotopic standards is not clear here, and is even less clear later when the -24 ‰ cylinder is dropped from the calibration routine.

Agreed. In retrospect that is clear. We added, "We note that it would have been preferable to utilize calibration tanks closer to the observed air samples in terms of isotopic ratio. In particular, the high tank could have been spiked with the –38.3 ‰ bottle, and/or both the high and the low tanks could have been spiked with a mixture of the –38.3 and –54.5 ‰ bottles. "

Page 9 Lines 11-13 – what is a large flow rate and what is the delay time for air entering the inlet to instrument measurement?

We added, "For the CRDS analyzer, air was drawn down the tube at 1 L/min, with 30 cc/min flow into the analyzer and the remainder purged. The residence time in the tube was about 1 min. Separate tubes were used for the CRDS and flask sampling lines because of the differing flow rates (varying between 0.29 and 3.8 liters per minute) required for the flask samples (Turnbull et. al., 2012) and to ensure independence of the CRDS and flask measurements. "

Page 9 Lines 28-30 – should the aim not be to achieve the best possible precision, not just to reach target compatibility? Would the high mole fraction tanks not achieve better precision if analysed for 32 minutes? If the shorter times are chosen just to save cal gas then it should be made clear somewhere.

We added to that paragraph, "The ideal calibration tank testing time is a balance between minimizing calibration gas usage (and consequently maximizing ambient air sampling time) and achieving sufficient precision. "

Page 10 Lines 18-24 – the -24 ‰ standard at any concentration was never going to be useful at the tower sites as it is even beyond the range of sampling a pure source emission. A clear understanding of the maximum mole fractions and isotopic shifts observed at the towers would have been useful before production of the working standards for these sites, even if it meant not using the first 6 months of isotope data.

Page 13 Lines 26-27 – what are the uncertainties on the NOAA tanks?

Uncertainties have been added to the text.

Page 14 Lines 8-10 – replace tubing with tube, and on-board with internal

Done.

Page 14 Section 5.3 – what is the relevance of this section? Unless the results of this test are shown somewhere, and referred to here, then is this section necessary for the points under discussion?

We added clarification that although not central to the primary results of this project, the performance of the analyzers is important if the data are to be used as part of the continental-scale CO2 network. The results are shown in Section 6.6.

Page 14 Line 16 – 14:00-15:00 isn't late afternoon

Replaced with "afternoon."

Page 14 Line 18 – do you mean measured? Either 'samples were collected' or 'flasks were filled'

Replaced with "samples were collected."

Page 15 Lines 9-10 – 'An error in isotopic ratio as a function of isotopic ratio' is not very informative. Figure 7 shows that there is an offset in the measured isotopic ratio as a function of the changing known isotopic ratio, which seems to be quite constant for all instruments at higher mole fractions, but instrument-specific at near-background mole fractions.

We added this clarification to the text.

Section 6.2 – I initially thought that the experiments involved changing over to different cylinders, which seems to be the case when changing to the new calibration routine, but most of these EXPTs seem to be just manipulation of the data for different standards, so please can you make clear at the start of this section that these are mostly changes in the calibration calculations and not changes to the cylinders being analysed. Some of the results of these experiments are not used as they do not improve the required precisions. Do these need to be described in the text?

This section was confusing and we have eliminated all but two schemes (using the target as independent and using the low as independent). We eliminated Table 3 from the original document. And we tried to clearly state when hindsight (and insightful reviewer comments) have revealed things that we would do differently.

Page 16 Line 13 – a result is a result and cannot be changed. Do you mean 'improve the calculated precision'?

Changed to "improve the calculated accuracy." Also replaced in three other locations in the text.

Page 18 Line 16 – 0.18 ‰ is stated above this as the daily average, not the hourly average as used here.

We clarified that section to indicate more clearly that averaging over the low tank for each day totaled about 1 hour of data (actually 81 min), and thus the standard deviation of these values is a proxy for the noise due to the calibration scheme in the hourly sample air data.

Section 6.6 – the side-by-side testing results are mentioned here, but have a lot of content overlap with the methodology for side-by-side testing. Does this have to be in two places? It isn't particularly relevant to the core isotopic story.

We were careful to keep the methods and results separate. We added clarification in Section 5.3 that although not central to the primary results of this project, the performance of the analyzers is important if the data are to be used as part of the continental-scale CO2 network.

Page 20 Line 2 – 'mean flask' – is this referring to the rapidly-filled or the hour-long filled flasks?

This was specified in the prior section as being the hour-long filled flasks.

Page 20 Line 5 – 'hourly flask to in-situ differences for the year'. It isn't clear what this means

This was repetitive with the prior section and makes it sound more complicated that it is.  We removed it.

Page 20 Line 19 – 'The time scale of the individual data points was 10 min'. Do you mean averaging interval?

Yes, corrected.

Page 20 Lines 32-33 – It isn't clear why 'enhancements greater than 6ppb CH4 in magnitude' are '3 times the target compatibility of 0.2 ‰

Typo corrected.

Page 21 Line 15 – mean of -31.2± ? ‰

-31.2 ± 1.9 ‰.  Added to text.

Page 21 Lines 21-23 – this section needs sorting out; firstly they are not peak heights but enhancements over background, and it is a 2.5 to 8.7 ‰ positive shift in measured isotopic ratio. What does ' reduced methane enhancement at other data points within the peak' mean? If the maximum enhancement is 2008 ppb, why does it mention 1500 ppb maximum earlier?

We made these changes and removed the confusing phrase as it is too obvious.  The maximum enhancement during non-afternoon hours at the central tower was 2008 ppb.  The 1500 ppb mentioned refers to afternoon hours at the North tower.  We added clarifications.

Page 22 Line 21 – remove the s from fractions

Done.

Page 22 Line 26 – replace improving with improved

Done.

Page 22 Lines 35 to 37 – given the availability of the Isometric flasks, my preference for a calibration would be to create a high and low mole fraction cylinder from both of the - 54 and -38 ‰ standards, used in combination with a low ambient cylinder and a similar target gas, or at least a high and low at -38 ‰ as that is the direction your sources are taking the ambient mix. More cylinders, but it should improve the correction in the triangle of 13C-mole fraction space where the measurements lie.

We have incorporated these ideas into Table 4 of the revised manuscript, for recommendations for future tower networks of CRDS isotopic methane analyzers.

Page 23 Line 30 – an isotopic ratio enhancement of -0.6 ‰ Two problems with this: surely this should be a + and not a -, and how do you enhance a ratio? Normally this would be heavy or light for a change in ratio as 13C increases or decreases, or enrichment or depletion if talking about the individual 13C, so it does represent an enrichment in 13C.

It seems that "difference from background isotopic ratio" is a better term than "enhancement above background" – I can see how the latter is confusing.  I switched the terminology throughout the text.  If the source in question is -35 ‰ and background is -47 ‰ (as in the example in the text), the measured isotopic ratio would be lower than background, so I think it is negative.  I'm assuming it was just the term "enhancement" that was causing the problem.

Page 24 Line 15 – already presented so should be ' we have presented.

Done.

Page 24 Lines 15-19 – this is a rather abrupt ending – see general comments for suggestions of what to add in summation.

References
All those present seem to be correct in the text, just move the date on P27 Line 7 up a line.

Done.

The following references are mentioned in the text but are not in the list: Conway et al., 2011 Montzka et al., 2011 Turnbull et al., 2012 Vaughn et al., 2004

These have been added to the references list.

Tables

Table 2 – Not clear what this table is representing from the caption? Are these due to interferences with the 12CH4 and 13CH4 spectral lines? What is the maximum CH4 at these sites and does this increase or decrease the interference on each of these species?

We clarified the caption of Table 2, "Maximum error estimate attributable to cross-interference due to direct absorption on $\delta$13CH4. These estimates were based on typical values for this tower-based application and estimated effects on CRDS measurements (Rella et al., 2015), and assumed 2 ppm ambient CH4 mole fraction. For water vapor and carbon dioxide, the interferences are independent of CH4 mole fraction for 1 – 15 ppm. For the other species listed, the interferences are inversely proportional to CH4 mole fraction. "

Table 3 caption Line 10 – should be at not and

Done.

Table 5 – the alternative strategy looks to be the best, but given the observed range of measurements up to 4 ppm, would it not be better to correct with a lower precision at 4 ppm than a calculation of the offset from 10 ppm?

This is a good question. The choice of the CH4 mole fraction of the high tank is based on the optimal determination of the calibration coefficients c0 and $\chi$, rather than the expected range of ambient CH4 mole fractions. The effect of the offset parameter c0 on the calibrated $\delta$ is largest at low mole fractions, whereas the effect of the slope parameter $\chi$ is independent of mole fraction. Thus the ratio of the high and low tank mole fractions determines how separable the two effects are. We therefore chose the high tank mole fraction to be as high as possible without introducing other nonlinearities into the system. We have added this explanation to the text.

Table 5 - You have all of the Isometric standards available, so 'easier to obtain tanks with these ratios' is not an advantage of the reduced compatibility option, but all of the options.

We have reworked Table 5 (now Table 4) to make it more clear.

Figures

Is there a way to sort out the superscript and subscripts on the vertical axes of the graphs or is this a software limitation?

The superscripts and subscripts have been fixed.

Figure 1 - It needs to be mentioned in the caption what are the signatures of the isotopic end members used to create this plot. There is an uneven spread of colour change around the -47 ‰ point, which suggests that this has been calculated with -30 and -60 ‰ source increments and not the -35 and -60 ‰ mentioned in the text.

Yes, thank you for pointing out this error. We have corrected the calculations for the figure to reflect -35 and -60‰, with a background of -47.5‰ used. We added further description of the end members to the caption, as requested.

Figure 2 caption – mention the cylinder volumes

Added: "At standard pressure and temperature, the gas volume of the zero air and working standard tanks was 4021 L and that of the Isometric Instruments bottles was 28 L."

Figure 4 – The lowest category of wind direction is 0-2 m/s, but the caption states that calm winds below 3.6 m/s are not categories by direction, so two things to correct here. 3.6 m/s is not calm. I think that this is a scale conversion problem from km/h, and calm should be either less than 1 m/s (in which case the lower category displayed should be 1-2 m/s), or <0.1 m/s.

3.6 m/s was a typo. Further clarification to the caption has been added. "These afternoon means were based on hourly reported measurements. For the hourly measurements, calm winds ($< 1.6$ m s$^{-1}$) were not categorized by direction and thus were not included in the afternoon mean. For the hourly measurements, calm winds ($< 1.6$ m s$^{-1}$) were reported as zero and were included in the afternoon mean. "

It isn't clear why Figs 3 and 4 are so early in the manuscript. Until the tower location data is discussed there isn't really the interest in seeing these influences.

We moved the section describing the study area (including the figures originally numbered 3 and 4) to just prior to the section describing "Methods for determining enhancements".

Figure 5 – What is the flow rate of the pump on the exhaust line (ME3)? It looks like it should be balancing out the ambient inlet flow.

Yes, the flow rate is about 1 LPM. We added that information to the figure, and also changed the labels of the tanks to be consistent with the text.

Figure 6 caption – incorrect 13C shown on first line

Corrected.

Figure 7 – The residual errors seem to be quite consistent between instruments for the higher mole fraction cylinders at all sites. The correction procedure for the small number of elevated mole fraction samples is not very obvious from the text.

The calibration procedure is described in Section 3.2.2.

Figures 8 and 9 – what are the error bars for these data points? If they cannot be added to the graphs can they be alluded to in the caption?

We added to the caption of Figure 8, "The Allan deviation for time period used for each calibration cycle was, for the period prior to the improved tank sampling strategy, 0.2 ‰ for the high tank, and 0.5 ‰ for the low and target tanks. Following the implementation of the improved tank sampling strategy, the Allan deviation for each calibration cycle was 0.1 ‰ for the high tank, and 0.3 ‰ for the low and target tanks. ", but it is not obvious how to calculate an estimation of the error for Figure 9.

Figure 11 – Can North, Central and East be labeled at the top of each column?

Done for this figure, and for Fig. 7.

Figure 13 – add measured before isotopic in the first line of the caption. The big delta small delta use on the left column horizontal axes is not explained. The x104 used in the right column needs correctly positioning, although it would be much clearer just using 0.2 to 0.6 in 1/CH4 ppm.

Done for this figure, and Figs. 14 and 15.

Figure 14 – the 15 ‰ spread of isotopic ratios measured at near background mole fractions reduces confidence in the data. The text suggests that there is a gradual improvement in measurement precision as the mole fraction increases. The Keeling plots suggest that the transition is quite sharp, and the isotope CRDS that I have seen in field operation seem to have a sharp improvement at 7-8 ppm CH4, so can this be clarified in the caption or elsewhere in the text.

Yes the range of values during non-afternoon hours at the North tower is 15 ‰, but the standard deviation is much less, only 0.76 ‰. The size of the markers in the figures makes it look more variable than it is. I reduced the marker size, which helped a little, but I don't think we can eliminate this without making the figures very large. Thus I added the median and standard deviation of the isotopic ratios at each tower to the Figs. 13/14 CFI, (Fig 14/15 in the updated document) in order to clarify the variability. The noise in the isotopic ratio measurement does increase near background mole fractions, but it's about 0.4 ‰ for 10 min averages at 2 ppm CH4. We added this to the captions for Figs. 13 and 14 and in the text. We added a figure showing the exponential decease of the standard deviation of the methane isotopic ratio at 2, 3, 6, and 7 ppm.

Figure 15 – the Keeling plots are good but indicate very little variation in measured isotopic ratio for a given mole fraction even at near background mole fractions. Are all of the points from these peaks used? It would be good to see the actual time graph of the isotopes for 1 of these graphs to see the points that have been selected for use in the Keeling plot. The source intercept calculation is without errors. Can these be calculated?

We added a figure to the Methods Section indicating an example of the $CH_4$ time series for which the Keeling plot approach was applied. The time during which the tower was in the plume of the source was obvious, and only these points were included in the calculation, thus explaining the lack of variability. Added to the text, "Propagating a potential error (attributable of analyzer uncertainty) of 0.2 ‰ at the heavy end of the Keeling plots and –0.2 ‰ at the light end, and vice versa, the potential range of the mean is from –32.0 to –30.4 ‰. "

**Anonymous Referee #2**

Review of the manuscript: "Calibration and Field Testing of Cavity Ring-Down Laser Spectrometers Measuring CH4, CO2, and δ13CH4 Deployed on Towers in the Marcel- lus Shale Region", submitted to Atmos. Meas. Tech. , by Natasha Miles et al.

The paper is describing the measurements of atmospheric mole fractions of δ13CH4, CH4 and CO2 at four sites in Pennsylvania. More precisely the manuscript describes the optimization of the technical setup based on lab and field tests.

The manuscript needs to be reorganized to reduce the back and forth between test descriptions and their results, which makes the reading quite difficult. There are too many redundancies, and unclear statement. When doing that I also suggest to shorten the manuscript. Some conclusions appear obvious, like for example the statement that field calibrations significantly improved the measurements compatibility. Also the so-called optimal calibration strategy refers to the design which was decided a priori and slightly modified during the campaign, but there was no plan to really evaluate alternative design. The conclusion should be written in a more concise way, focusing on the recommendations gained from the experiment.

Thank you for your thoughtful comments. We have addressed these concerns by reorganizing the bulk of the paper. Instead of strictly separating methods and results, we have now inter-mixed these, with the paper being organized by topic. I believe this does enhance readability of the paper. We shortened the paper by removing most of the discussion of choosing the optimal calibration study. As the other reviewer indicated, the isotopic ratio of the low tank (-23.9 per mil) would be better chosen to be closer to the measured values. We also eliminated switching from the target tank being independent to the low tank being independent in the text. Instead we just described how we actually processed the data, rather than describing all the options we explored. We have reworked the conclusions, focusing on the recommendations gained from our experiment and the potential of high-temporal-resolution isotopic methane data to constrain regional methane budgets.

Introduction: the introduction need to be reorder in order. For example the first paragraph of page 4 describing the interest of tower versus aircraft, appears between two paragraphs discussing more technical points about CRDS measurements

We reordered the introduction and added context concerning the utility of high-temporal-resolution isotopic methane data.

Page 4 / Line 13: "three field calibration tanks. . .": I would rather say two calibration tanks plus one target tank used as quality control and not used in the calibration.

We changed this throughout the text to refer to these tanks as field tanks, since as you mention, one of them is independent of the calibration.

Allan variances tests; calibrations tests (Page 6 / Line 31): there are many back and forth between description of the set up and the results, which confuse the paper.

We have reorganized the paper to eliminate this concern. Instead of strictly separating methods and results, we have now inter-mixed these, with the paper being organized by topic. We believe this does enhance readability of the paper. While not the traditional method of manuscript organization, it is sometimes used (e.g., Rella et al. 2015).

Page 9: In-situ field calibration: is the Nafion required for the setup ? Have you compared possible biases due to the use of the Nafion versus the water vapor correction ? I am not fully convinced by the strategy of humidifying the dry calibration tanks.

Rella et al. (2015) noted that the effect of water vapor on the isotopic ratio of methane is large (up to 1 ‰) and nonlinear. Thus no water vapor correction is available. They recommend drying to less than 0.1% mole fraction. We have added clarification on our reasoning for drying to the text. We also added a reference to Andrews et al. (2014) who document the technique of drying the sample and humidifying the calibration gases.

Page 9: 4 min flushing: how do you estimate those 4 minutes as sufficient for the flushing ?

Added to the text: After this time, the $CO_2$ and $CH_4$ mole fractions have stabilized.

Page 12: background site: why don't you select the background site as a function of wind direction rather than picking up one site for the full period ?

We added to this discussion. It now reads, ". The predominant wind direction for the Marcellus region is from the west (Fig. 4). For westerly winds, the South tower is a reasonable choice for a background tower. The South tower measured the lowest overall mean afternoon methane mole fraction (1960.2 ppb $CH_4$). The mean afternoon methane mole fractions of the other towers, averaged only when data for the South tower exist, were 8.7, 7.0, and 2.9 ppb higher, at the North, Central, and East towers, respectively. For future analyzes, a wind direction dependent background tower (South or North) could be used, but the North tower did have the largest mean enhancement in $CH_4$ mole fraction compared to the South tower. "

Page 14: Allan results: For $CH_4$ and $CO_2$ it should be noted that the results seem to be not as good as the performances obtained with G1301/G2401 analyzers. Do you know the reason which could explain a difference of the performances between those analyzers ?

We have added further clarification to the Allan deviation results section and the side-by-side testing section. The performance of the G2132-i analyzers in terms of $CO_2$ precision is worse than that of the G2301/G2401 analyzers primarily because a weaker spectral line is used. Whereas the spectral line for $CH_4$ is the same between the two types of analyzers, for $CO_2$, the absorbance of the spectral line used in the G2132-i analyzers is a factor of 11 times less, meaning the precision is dramatically reduced.

Page 16: Calibration scheme: the presentation of the different tests should probably be shortened. Is there a difference between Expt E and H designs ?

We have eliminated most of the EXPTs in order to simplify. We eliminated Table 3 and shortened Table 4 from the original document. EXPTs E and H are no longer described.

I would appreciate an evaluation of the optimum frequency of the field calibration sequences (intermediate between 0 and once per day). From the variabilities shown on Fig.8 and 9 it looks like a reduction of the calibration frequency to once every few days would not affect by much the measurements.

We added to the text, 'Considering shorter term changes, the day to day changes in the calibration were less than 0.5 ‰ for December 2016. Less frequent calibrations, e.g., twice per week, could be considered, but the reduction in field tank use is not

large considering the low flow rates of the instruments and steady changes up to 2 ‰ in the raw data over the time scale of days were observed in Rella et al. (2015). '

Fig.8: the legend is misleading since the so-called target tank is used as a calibration tank. To make it clear you should add comments in the legend of each figure (e.g. Target tank (used as CAL))

We added to the caption, "The target tank was used in the isotopic ratio calibration, whereas the low tank was independent."

Page 17: Fig. 9B and 9C should rather be 9A and 9B

Corrected.

Page 20 Line 27: suppress 'For the daily afternoon averages,'. Not clear what you mean by a 'reduction' of 0.6-0.7pmil.

We have clarified these confusing statements.  The text now reads,  "The standard deviation of the daily afternoon averages (rather than 10-min averages) was 0.6 – 0.7 ‰. Thus the observed width of the distribution appears to be persistent throughout the afternoon and not merely measurement noise. "

Page 20 Line 32: Why do you compare CH4 enhancements (6ppb) with 13CH4 target compatibility (0.2pmil) ?

Typo corrected.

Page 21: lines 22/23: Unclear statement about the dilution of local source.

We have removed this statement.

Page 21: lines 22/23: The discussion about the source signature need to be clarified, or preferably merged in the discussion section.

Hopefully having the paper organized by topic, rather than having the method for each topic separated from the results, makes this discussion more clear.  We also added a figure of an example time series of a peak for which we applied this method.

Page 23: lines21/22: unclear statement.

A misplaced parenthesis made this statement confusing.  Corrected.

Conclusion: in your last sentence I would like to see also a comment or discussion that the strategy of using continuous measurements at four tower is maybe not the optimal one for the quantification of such sources.

We added, ' For determination of the source signature for a specific known location, the tower-based approach is not ideal. Instead the strength of the tower-based approach lies in covering larger areas and many potential source locations, and for longer periods of time than is feasible by other approaches, e.g., short-term mobile techniques. ' We also added to the last section discussion about the utility of high-temporal-resolution methane isotopic ratio data for constraining regional methane budgets.

Please also note the supplement to this comment: https://www.atmos-meas-tech-discuss.net/amt-2017-364/amt-2017-364-RC2-supplement.pdf

[revised manuscript text omitted]

Moved (insertion) [11]

Moved up [14]: .

Formatted [...23]

Moved up [4]: The resulting Allan standard deviation [...25]

Moved up [5]: . For the high tank, the Allan deviatio [...26]

Moved up [23]: The analyzer deployed at the South t [...12]

Moved up [24]: Four to six tests were completed for [...14]

Moved up [27]: analyzers were run side-by-side for c [...16]

Moved up [18]: ) in Eq. (3) are time-dependent drift [...34]

Moved up [31]: ), halo- and hydro-carbons (Montzka [...18]

Moved up [32]: afternoon (1400–1500 LST), thereb [...20]

Moved up [6]: ) was < 0.2 ‰ (our target compatibilit [...27]

Moved up [8]: The $CO_2$ levels in the high and low ta [...29]

Moved up [15]: for the period September – Decemb [...31]

Moved up [16]: . [...33]

Moved up [17]: . The terms $c_0$ (Fig.

Moved up [19]: The low tank differences from know [...36]

Moved up [20]: During this period, the calibration us [...38]

Moved up [21]: Based on this result, differences in [...40]

Moved up [22]: Therefore, according to this metric, a [...42]

Moved up [25]: .

Formatted [...44]

Moved up [26]: 229, 2016). The standard error, ind [...46]

Moved up [28]: resulted in mean differences of 0.06 [...48]

Moved up [29]: The G2132-i analyzers are thus [...50]

Moved up [30]: To optimize results on a daily time s [...51]

Moved up [33]: For January – December 2016, the n [...52]

[revised manuscript text omitted]

An alternative calibration approach is

| Page 4: [2] Deleted | Microsoft Office User | 1/24/18 10:53:00 AM |

2017).

Here we

| Page 4: [3] Moved to page 4 (Move #2) | Microsoft Office User | 1/24/18 10:53:00 AM |

In this paper, we describe a network of four tower-based atmospheric observation locations, measuring $CH_4$ and $CO_2$ dry mole fractions and $\delta^{13}CH_4$ using CRDS (Picarro, Inc., model G2132-i) analyzers in the Marcellus shale region in north-central Pennsylvania

| Page 5: [4] Deleted | Microsoft Office User | 1/24/18 10:53:00 AM |

**3 Methods: Laboratory testing**

**3.1**

| Page 8: [5] Moved to page -570016888 (Move #10) | Microsoft Office User | 1/24/18 10:53:00 AM |

**.1 Study area**

**Four CRDS isotopic $CH_4$ analyzers (G2132-i, Picarro, Inc.) were deployed on commercial towers 46–61 m AGL in northeast Pennsylvania (Fig.**

| Page 8: [6] Deleted | Microsoft Office User | 1/24/18 10:53:00 AM |

2012), measuring a suite of $> 55$ gases (including greenhouse gases, hydrocarbons, and halocarbons) and $\delta^{13}CH_4$.

**4.2**

| Page 9: [7] Deleted | Microsoft Office User | 1/24/18 10:53:00 AM |

The sampling scheme and procedure for using these field calibration tanks at each tower to correct the ambient $\delta^{13}CH_4$ measurements is described in Sections 3.2.2 and 6.2.

**4.3**

| Page 17: [8] Moved from page 8 (Move #10) | Microsoft Office User | 1/24/18 10:53:00 AM |

**.1 Study area**

Four CRDS isotopic $CH_4$ analyzers (G2132-i, Picarro, Inc.) were deployed on commercial towers 46–61 m AGL in northeast Pennsylvania (Fig.

| Page 17: [9] Moved from page 8 (Move #11) | Microsoft Office User | 1/24/18 10:53:00 AM |

). The South and North towers were located on the southern and northern edges of the unconventional gas well region, respectively, and were intended to measure background values depending on the wind direction.

| Page 17: [9] Moved from page 8 (Move #11) | Microsoft Office User | 1/24/18 10:53:00 AM |

| Page 17: [10] Deleted | Microsoft Office User | 1/24/18 10:53:00 AM |

| Page 17: [10] Deleted | Microsoft Office User | 1/24/18 10:53:00 AM |

| Page 17: [10] Deleted | Microsoft Office User | 1/24/18 10:53:00 AM |

| Page 17: [10] Deleted | Microsoft Office User | 1/24/18 10:53:00 AM |

**Page 17: [11] Deleted**         **Microsoft Office User**         **1/24/18 10:53:00 AM**

Keeling plots (Keeling 1961; Röckmann et al., 2016) are often used to infer the isotopic ratio of the methane source as the intercept of the best fit line of the isotopic ratio as a function of the inverse methane mole fraction. In Section 6.7 we used this approach to estimate the source isotopic ratio of peaks observed during non-afternoon hours at the Central tower.

**5 Methods for evaluating compatibility of in-situ tower measurements**

**5.1 Independent low tank**

While the low tank was planned to be used in the calibration of the isotopic ratio, the optimized calibration scheme given the deployed tanks instead utilized the target tank in the calibration and kept the low tank as independent (Section 6.2). The low tank was thus treated as an ambient sample. To evaluate the noise in the calibrated ambient samples that results from noise in the calibration, we calculated the standard deviation over the period September 1 – December 2 of the individual calibration cycles (6 min each), of the calibration cycles averaged over 1 day (81 min total), and of the calibration cycles averaged over 3 days (4.1 hours total). These results are a proxy for the noise in the calibrated ambient samples over those sampling periods. The same calculation was performed for the period December 3 – December 31, a period during which an improved calibration tank sampling scheme was utilized.

**5.2 Round-robin testing**

Post-deployment round-robin style tests were completed in the laboratory in March 2017 for the analyzers previously deployed at the North, Central and East Towers, in order to assess the compatibility achievable via our calibration method.

**Page 17: [12] Moved to page 14 (Move #23)**         **Microsoft Office User**         **1/24/18 10:53:00 AM**

The analyzer deployed at the South tower was not included in these tests, as it was still in the field. Two NOAA/INSTAAR tanks (JB03428: –46.82 ‰ $\delta^{13}CH_4$, 1895.3 ppb $CH_4$ and 381.63 ppm $CO_2$; and JB03412: –45.29 ‰ $\delta^{13}CH_4$, 2385.2 ppb $CH_4$ and 432.71 ppm $CO_2$) were

**Page 17: [13] Deleted**         **Microsoft Office User**         **1/24/18 10:53:00 AM**

sampled for 70 min, with 8 min ignored after each transition, and treated as unknowns. Additionally, high, low, and target tanks were sampled, with the calibration applied as in the field for ambient samples (as described in Section 6.2). The high mole fraction tank was sampled for 20 min and the low and target mole fraction tanks were sampled for 70 min, with 8 min ignored after each gas transition.

**Page 17: [14] Moved to page 14 (Move #24)**         **Microsoft Office User**         **1/24/18 10:53:00 AM**

Four to six tests were completed for each analyzer. We used these tests as a means of evaluating the compatibility of the analyzers, in terms of both mole fractions and the isotopic ratio.

| Page 17: [15] Deleted | Microsoft Office User | 1/24/18 10:53:00 AM |

**5.3 Side-by-side testing**

The precision and drift characteristics are not optimized for $CO_2$ for the G2132-i analyzers, compared to the G2301 analyzers, which measure $CO_2$, $CH_4$ and $H_2O$ mole fractions. To test the performance of the G2132-i analyzers for consideration of the data for use as part of the continental-scale $CO_2$ network, G2301 and G2132-i (Picarro, Inc.)

| Page 17: [16] Moved to page 15 (Move #27) | Microsoft Office User | 1/24/18 10:53:00 AM |

analyzers were run side-by-side for one month (June 2016) at the South tower. The sampling system for the G2132-i was as described in Section

| Page 17: [17] Deleted | Microsoft Office User | 1/24/18 10:53:00 AM |

4.3. A separate ¼" (0.64 cm) tubing was used for the G2301 analyzer and an intercept calibration using the target tank is applied daily. The sample air for the G2301 analyzer was not dried and the on-board water vapor correction was used. This testing was used to evaluate the mole fraction compatibility, particularly for $CO_2$, of the G2132-i analyzers compared to the G2301 analyzers.

**5.4 Flask measurements**

Flask measurements were used for independent validation and error estimation of the continuous $CO_2$, $CH_4$ and $\delta^{13}CH_4$ in-situ measurements. In addition, the flasks were measured for a suite of species including $N_2O$, $SF_6$, CO, $H_2$ (Conway et al., 2011

| Page 17: [18] Moved to page 16 (Move #31) | Microsoft Office User | 1/24/18 10:53:00 AM |

), halo- and hydro-carbons (Montzka et al., 1993) and stable isotopes of $CH_4$ (Vaughn et al., 2004).

| Page 17: [19] Deleted | Microsoft Office User | 1/24/18 10:53:00 AM |

The flasks were filled over a 1-hour time period in the late

| Page 17: [20] Moved to page 16 (Move #32) | Microsoft Office User | 1/24/18 10:53:00 AM |

afternoon (1400–1500 LST), thereby yielding a more representative measurement compared to most flask sampling systems, which collect nearly instantaneous samples (e.g., ~10 sec).

| Page 17: [21] Deleted | Microsoft Office User | 1/24/18 10:53:00 AM |

Samples were measured only when winds were blowing steadily out of the west or north (~45–225°) to ensure that the samples are sensitive to and representative of the broader Marcellus shale gas production region that is the focus of this study.

**6 Results**

6.1 Allan standard deviation results

| Page 17: [22] Moved to page 13 (Move #14) | Microsoft Office User | 1/24/18 10:53:00 AM |

As

| Page 17: [23] Formatted | Microsoft Office User | 1/24/18 10:53:00 AM |

Font color: Auto, English (US)

| Page 17: [24] Deleted | Microsoft Office User | 1/24/18 10:53:00 AM |

described in Section 3.1, two tanks were sampled for 24 hours each to determine the Allan standard deviation as a function of averaging interval for the G2132-i analyzers.

| Page 17: [25] Moved to page 6 (Move #4) | Microsoft Office User | 1/24/18 10:53:00 AM |

The resulting Allan standard deviations for $\delta^{13}CH_4$, $CH_4$ and $CO_2$ are shown in Fig.

| Page 17: [26] Moved to page 6 (Move #5) | Microsoft Office User | 1/24/18 10:53:00 AM |

. For the high tank, the Allan deviation for $\delta^{13}CH_4$ (Fig.

| Page 17: [27] Moved to page 6 (Move #6) | Microsoft Office User | 1/24/18 10:53:00 AM |

) was < 0.2 ‰ (our target compatibility) for an averaging interval of 2 min (the averaging interval used each field calibration cycle of the high tank). To reduce the noise to < 0.1 ‰, an averaging interval of 4 min is sufficient (in addition to the time required for the transition between gases). For the low tank, in order for the Allan standard deviation to be < 0.2 ‰, 32 min were required and 64 min for 0.1 ‰ noise.

[revised manuscript text omitted]

**Page 17: [44] Formatted**       **Microsoft Office User**       **1/24/18 10:53:00 AM**

Font color: Auto, English (US)

**Page 17: [45] Deleted**       **Microsoft Office User**       **1/24/18 10:53:00 AM**

of two NOAA/INSTAAR tanks are shown in Fig. 11.  The mean of the errors (measured – NOAA known value) calculated from the results of four to six tests for each analyzer were –0.08 to 0.04 ppm $CO_2$ within the 0.1 ppm WMO compatibility recommendation for global studies of $CO_2$ (GAW Report No.

**Page 17: [46] Moved to page 14 (Move #26)**       **Microsoft Office User**       **1/24/18 10:53:00 AM**

 229, 2016).  The standard error, indicating an estimate of how far the sample mean is likely to be from the true mean, for the means of the $CO_2$ tests were 0.03 – 0.10 ppm.  The mean difference was –0.03 to 0.02 ppm $CO_2$ for the analyzers, averaged over the two round-robin tanks (analogous to averaging over the entire range of $CO_2$ during the flask comparison, for example). For $CH_4$, the means of the errors were 0.03 – 0.07 ppb $CH_4$, for the NOAA/INSTAAR tank measuring 2385.2 ppb, and –0.83 to – 0.70 ppb $CH_4$ for the NOAA/INSTAAR tank measuring 1895.3 ppb $CH_4$. Therefore, there was a slight error in the slope of the linear calibration, possibly attributable to tank assignment errors.

However, the error was well within the WMO recommendations for global studies of 2 ppb $CH_4$ (GAW Report No. 229, 2016), and the range of NOAA/INSTAAR tanks encompassed the majority of the $CH_4$ mole fraction observed during the study. We also note that the standard error for the means of the $CH_4$ tests were $0.07 - 0.12$ ppb. Averaging over the two round-robin tanks, the mean difference was $-0.40$ to $-0.32$ ppm $CH_4$ for the analyzers. For $\delta^{13}CH_4$, the mean errors for each analyzer/tank pair were $-0.33$ to $0.24$ ‰ for these tanks within the range of ambient isotopic ratio and the standard errors were $0.05 - 0.10$ ‰. The mean errors were $-0.14$ to $0.03$ ‰ for each analyzer.

| Page 17: [47] Deleted | Microsoft Office User | 1/24/18 10:53:00 AM |
|---|---|---|

**6.5 Side-by-side testing**

Side-by-side testing of a G2301 ($CO_2/CH_4/H_2O$) analyzer and a G2132-i analyzer ($CH_4/\delta^{13}CH_4/CO_2$) for June 2016 at the South tower

| Page 17: [48] Moved to page 15 (Move #28) | Microsoft Office User | 1/24/18 10:53:00 AM |
|---|---|---|

resulted in mean differences of $0.06\pm0.41$ ppm $CO_2$ and $0.9\pm1.5$ ppb $CH_4$, with the G2132-i analyzer measuring slightly lower for both species. Here the standard deviation was based on the 10-min average calibrated values for the month for all times of the day. The standard error of the differences was $0.01$ ppm $CO_2$ and $0.02$ ppb $CH_4$. These results indicate that the performance of the G2132-i is similar for $CO_2$ and $CH_4$ mole fractions, at least in terms of the long-term mean. In terms of utilizing the mole fraction data in atmospheric inversions, the multi-day mean afternoon differences are

| Page 17: [49] Deleted | Microsoft Office User | 1/24/18 10:53:00 AM |
|---|---|---|

more appropriate. The five-day mean afternoon difference for the month was $0.05\pm0.08$ ppm $CO_2$ and $-0.7\pm0.1$ ppm $CH_4$.

[revised manuscript text omitted]

Formatted Table

| Page 31: [72] Formatted | Microsoft Office User | 1/24/18 10:53:00 AM |
|---|---|---|

Font color: Text 1, Pattern: Clear (White)

| Page 31: [73] Formatted | Microsoft Office User | 1/24/18 10:53:00 AM |
|---|---|---|

Font color: Text 1

| Page 31: [74] Formatted | Microsoft Office User | 1/24/18 10:53:00 AM |
|---|---|---|

Font color: Text 1, Pattern: Clear (White)

| Page 31: [75] Formatted | Microsoft Office User | 1/24/18 10:53:00 AM |
|---|---|---|

Font color: Text 1

| Page 31: [76] Formatted | Microsoft Office User | 1/24/18 10:53:00 AM |
|---|---|---|

Font color: Text 1, Pattern: Clear (White)

| Page 31: [77] Formatted | Microsoft Office User | 1/24/18 10:53:00 AM |
|---|---|---|

Font color: Text 1

| Page 31: [78] Formatted | Microsoft Office User | 1/24/18 10:53:00 AM |
|---|---|---|

Font color: Text 1, Pattern: Clear (White)

| Page 31: [79] Formatted | Microsoft Office User | 1/24/18 10:53:00 AM |
|---|---|---|

Pattern: Clear (White)

| Page 31: [80] Formatted | Microsoft Office User | 1/24/18 10:53:00 AM |
|---|---|---|

Font color: Text 1

| Page 31: [81] Formatted | Microsoft Office User | 1/24/18 10:53:00 AM |
|---|---|---|

Font color: Text 1

| Page 31: [82] Formatted | Microsoft Office User | 1/24/18 10:53:00 AM |
|---|---|---|

Font color: Text 1

| Page 31: [83] Formatted | Microsoft Office User | 1/24/18 10:53:00 AM |
|---|---|---|

Font color: Text 1

| Page 31: [84] Formatted | Microsoft Office User | 1/24/18 10:53:00 AM |
|---|---|---|

Font color: Text 1

| Page 31: [85] Formatted | Microsoft Office User | 1/24/18 10:53:00 AM |
|---|---|---|

Font color: Text 1

| Page 31: [86] Formatted | Microsoft Office User | 1/24/18 10:53:00 AM |
|---|---|---|

Font color: Text 1

| Page 31: [87] Formatted | Microsoft Office User | 1/24/18 10:53:00 AM |
|---|---|---|

Font color: Text 1

| Page 31: [88] Formatted | Microsoft Office User | 1/24/18 10:53:00 AM |
|---|---|---|

Font color: Text 1

| Page 31: [89] Formatted | Microsoft Office User | 1/24/18 10:53:00 AM |
|---|---|---|

Font color: Text 1

| Page 31: [90] Formatted | Microsoft Office User | 1/24/18 10:53:00 AM |
|---|---|---|

Font color: Text 1

| Page 31: [91] Formatted | Microsoft Office User | 1/24/18 10:53:00 AM |
|---|---|---|

Font color: Text 1

| Page 31: [92] Formatted | Microsoft Office User | 1/24/18 10:53:00 AM |
|---|---|---|

Font color: Text 1

| Page 31: [93] Formatted | Microsoft Office User | 1/24/18 10:53:00 AM |

Font color: Text 1

| Page 31: [94] Formatted | Microsoft Office User | 1/24/18 10:53:00 AM |

Font color: Text 1

| Page 31: [95] Formatted | Microsoft Office User | 1/24/18 10:53:00 AM |

Font color: Text 1

| Page 31: [96] Formatted | Microsoft Office User | 1/24/18 10:53:00 AM |

Font color: Text 1

| Page 31: [97] Formatted | Microsoft Office User | 1/24/18 10:53:00 AM |

Font color: Text 1

| Page 31: [98] Formatted | Microsoft Office User | 1/24/18 10:53:00 AM |

Font color: Text 1

| Page 31: [99] Formatted | Microsoft Office User | 1/24/18 10:53:00 AM |

Font color: Text 1

| Page 31: [100] Formatted | Microsoft Office User | 1/24/18 10:53:00 AM |

Font color: Text 1

| Page 32: [101] Deleted | Microsoft Office User | 1/24/18 10:53:00 AM |

HIGH (10 ppm,

−38.3‰, 4 min/day)

| Page 32: [102] Deleted | Microsoft Office User | 1/24/18 10:53:00 AM |

LOW (2 ppm,

−23.9‰, 6–32 min/day)

| Page 35: [103] Deleted | Microsoft Office User | 1/24/18 10:53:00 AM |

6 port dead-end,
common outlet
flow path selector
(Valco)

0.005-1 sccm
Range MFC

130 sccm

2500 ppm CH$_4$
$\delta^{13}$-23.9 ‰
(Isometric)

2500 ppm CH$_4$
$\delta^{13}$-38.3 ‰
(Isometric)

Mixing
Volume

2500 ppm CH$_4$
$\delta^{13}$-66.5 ‰
(Isometric)

1-200 sccm
Range MFC

2500 ppm CH$_4$
$\delta^{13}$-54.5 ‰
(Isometric)

Outlet Pressure ~ 4 psi

Zero Air
(Scott-Marrin)

Working
Standard

30 sccm

30 sccm

30 sccm

NOTE: Outlet Pressure on Isometric and Scott-Marrin tanks
set based on inlet specifications of MFCs

CRDS

CRDS

CRDS

CRDS

Pump

Pump

Pump

Pump

30 sccm

~10 sccm

[Figure]

Page 39: [105] Deleted   Microsoft Office User   1/24/18 10:53:00 AM

[Figure]

Map of Pennsylvania with permitted unconventional natural gas wells (magenta dots) and network of towers with methane and stable isotope analyzers (Picarro G2132-i).  The East and South towers were also equipped with NOAA flask sampling systems.  The Binghamton Airport is also indicated.

[Figure]

**Figure 5.**

m s$^{-1}$) are not categorized by direction.

[Figure]

**Figure 6**

Allan standard deviation for (A) $\delta^{13}CH_4$, (B) $CH_4$, and (C) $CO_2$ for a high $CH_4$ mole fraction tank (9.7 ppm $CH_4$, ~400 ppm $CO_2$, −38.3 ‰ $\delta^{13}CH_4$) (orange) and a low (1.9 ppm $CH_4$, ~400 ppm $CO_2$, −23.7 ‰ $\delta^{13}CH_4$) tank (blue). The x-axis is truncated to focus on minimum averaging times required to achieve the desired compatibility goals.

[Figure]

**Figure 8.**

[Figure]

Page 42: [112] Deleted | Microsoft Office User | 1/24/18 10:53:00 AM

[Figure]

Page 44: [113] Deleted                    Microsoft Office User                    1/24/18 10:53:00 AM

[Figure]

The standard deviation of the in-situ to flask differences are shown in parentheses on each plot. The standard errors, indicating an estimate of how far the sample mean is likely to be from the true mean, is 0.24 ppb $CH_4$, 0.03 ppm $CO_2$ and 0.06 ‰ at the East tower and 0.14 ppb $CH_4$, 0.04 ppm $CO_2$ and 0.04 ‰ at the South tower.